# Prediction error signaling explains neuronal mismatch responses in the medial prefrontal cortex

**Lorena Casado-Román**[1,2⊙], **Guillermo V. Carbajal**[1,2⊙], **David Pérez-González**[1,2]*, **Manuel S. Malmierca**[1,2,3]*

**1** Cognitive and Auditory Neuroscience Laboratory (CANELAB), Institute of Neuroscience of Castilla y León (INCYL), Salamanca, Spain, **2** Institute for Biomedical Research of Salamanca (IBSAL), Salamanca, Spain, **3** Department of Biology and Pathology, Faculty of Medicine, University of Salamanca, Salamanca, Spain

⊙ These authors contributed equally to this work.
* davidpg@usal.es (DPG); msm@usal.es (MSM)

**Data Availability Statement:** All relevant data are within the paper and its Supporting Information files.

## Abstract

The mismatch negativity (MMN) is a key biomarker of automatic deviance detection thought to emerge from 2 cortical sources. First, the auditory cortex (AC) encodes spectral regularities and reports frequency-specific deviances. Then, more abstract representations in the prefrontal cortex (PFC) allow to detect contextual changes of potential behavioral relevance. However, the precise location and time asynchronies between neuronal correlates underlying this frontotemporal network remain unclear and elusive. Our study presented auditory oddball paradigms along with "no-repetition" controls to record mismatch responses in neuronal spiking activity and local field potentials at the rat medial PFC. Whereas mismatch responses in the auditory system are mainly induced by stimulus-dependent effects, we found that auditory responsiveness in the PFC was driven by unpredictability, yielding context-dependent, comparatively delayed, more robust and longer-lasting mismatch responses mostly comprised of prediction error signaling activity. This characteristically different composition discarded that mismatch responses in the PFC could be simply inherited or amplified downstream from the auditory system. Conversely, it is more plausible for the PFC to exert top-down influences on the AC, since the PFC exhibited flexible and potent predictive processing, capable of suppressing redundant input more efficiently than the AC. Remarkably, the time course of the mismatch responses we observed in the spiking activity and local field potentials of the AC and the PFC combined coincided with the time course of the large-scale MMN-like signals reported in the rat brain, thereby linking the microscopic, mesoscopic, and macroscopic levels of automatic deviance detection.

## Introduction

Since the discovery of the mismatch negativity (MMN) 4 decades ago [1,2], this biomarker has become a pivotal tool for cognitive and clinical research in the human brain [3,4], even showing potential diagnostic capabilities [5]. The MMN reflects how the nervous system

**Funding:** This project has received funding from the European Union's Horizon 2020 research and innovation programme [Marie Skłodowska-Curie Grant agreement No. 722098 (LISTEN)] to MSM; Spanish AEI (PID2019-104570RB-I00) to MSM. DPG held a Postdoctoral salary from the Spanish Ministry of Science and Innovation (MICINN, Grant No. SAF2016-75803-P). Fellowship from the European Union's Horizon 2020 research and innovation programme [Marie Skłodowska-Curie Grant agreement No. 722098 (LISTEN)] to LCR. Fellowship from the Spanish MICINN (BES-2017-080030) to GVC. The funders had no role in study design, data collection and analysis, decision to publish, or preparation of the manuscript.

**Competing interests:** I have read the journal's policy and the authors of this manuscript have the following competing interests. Manuel S. Malmierca is an Academic Editor for PLOS Biology. The other authors have declared that no competing interests exist.

**Abbreviations:** AC, auditory cortex; ACC, anterior cingulate cortex; CTR, control condition; dB SPL, decibels of sound pressure level; DEV, deviant condition; ECoG, electrocorticography; ERP, event-related potential; FDR, false discovery rate; FRA, frequency response area; IC, inferior colliculus; IL, infralimbic cortex; iMM, index of neuronal mismatch; iPE, index of prediction error; iRS, index of repetition suppression; LFP, local field potential; MGB, medial geniculate body; mPFC, medial prefrontal cortex; MMN, mismatch negativity; M2, secondary motor cortex; PE, prediction error; PE-LFP, prediction error potential; PFC, prefrontal cortex; PL, prelimbic cortex; SEM, standard error of the mean; SSA, stimulus-specific adaptation; STD, standard condition.

automatically encodes regular patterns in the sensorium, generates internal models to explain away those regularities, and detects deviations from those internal representations in upcoming sensory input, a processing mechanism that is key for survival [6]. This automatic process of deviance detection is commonly studied using an oddball paradigm, where a sequence of repetitive "standard" tones is randomly interrupted by another rare "deviant" tone. When the scalp-recorded auditory event-related potential (ERP) elicited by a tone presented in the standard condition (STD) is subtracted from the ERP prompted by that same tone presented in the deviant condition (DEV), a "mismatch" response (*DEV–STD*) becomes visible at temporal and frontal electrodes in the form of a slow negative deflection; hence the name mismatch negativity [1,2,6].

The topographic distribution of the MMN reveals a frontotemporal network in charge of automatic deviance detection [7–9]. According to the classic cognitive interpretation of the MMN [4,10], temporal sources from the auditory cortex (AC) would first encode acoustic regularities in a sensory memory, detecting specific sensory deviances between that memory trace and incoming input [11]. Then, additional sources from the prefrontal cortex (PFC) assess the behavioral relevance of that sensory deviance, potentially triggering an attention switch toward the change [12–14]. A more neurophysiologically grounded interpretation of the MMN, known as the adaptation hypothesis, denies the existence of a genuine process of deviance detection, arguing that the STD induces stimulus-specific adaptation (SSA) on AC neurons [15,16], whose frequency channels simply remain fresh to keep responding to the DEV [17,18]. Despite their conceptual disparities, both the sensory-memory and the adaptation hypotheses agree that early AC processing is highly sensitive to specific stimulus features. Conversely, PFC activity seems more reliant on an overall evaluation of global properties, which occurs upstream of initial sensory discrimination processes [6,19].

Recent proposals under the predictive processing framework have attempted to integrate previous accounts of the generation of the MMN (for a recent in-depth discussion, see [20]), establishing a hierarchical and reciprocal relationship between the AC and the PFC. The AC would first represent the spectral properties of sensory stimuli, suppressing redundant auditory inputs based on their frequency-specific features, by means of short-term plasticity mechanisms such as synaptic depression and lateral inhibition [21–23]. During an oddball paradigm, this would be functionally observable as SSA, or more appropriately, as repetition suppression [22,24–26]. The information that could not be explained away in the AC is forwarded as a prediction error signal (PE) to higher levels in the processing hierarchy [27,28]. Eventually, the bottom-up flow of PEs reaches the PFC, which tries to explain PEs away by means of higher-order expectations regarding emergent properties of the auditory stimulation, such as complex interstimulus relationships and structures [22,29,30]. Thus, whereas fast PEs forwarded from the AC are purely auditory in nature, the PFC would generate PEs when more abstract expectations are not met, requiring an update.

Despite the several hypotheses accounting for MMN generation, its neuronal substrate remains elusive and poorly understood, mostly due to the ethical constraints on human brain research. Noninvasive techniques, such as ERP analysis or functional magnetic resonance imaging, cannot pinpoint response measurements with enough temporal and spatial resolution as to deem with absolute certainty whether AC potentials precede those from the PFC [31–33]. When invasive approaches are available, electrocorticography (ECoG) electrode placement in human patients is strictly restrained by clinical criteria, causing intra- and interindividual variability that hampers systematic and detailed comparisons [34–37]. In contrast, invasive techniques of electrophysiological recording in animal models offer both the spatial and temporal resolution necessary to compare mismatch signals across areas more precisely. Auditory-evoked spiking activity and local field potentials (LFPs) can provide the accurate

locations and time courses of mismatch responses at microscopic and mesoscopic levels, respectively [38,39]. In turn, those local-scale mismatch responses can be correlated with the large-scale MMN-like potentials which are thought to be the specific analog of the human MMN in the corresponding animal model [40,41]. Hence, animal models can help to define the neuronal substrate of the human MMN, as well as to ratify or discard certain hypotheses about its generation.

In the present study, we recorded spiking activity and LFPs from 1 possible frontal source contributing to the emergence of MMN-like potentials in the rat brain: the medial prefrontal cortex (mPFC). Following the standards of the most thorough human MMN studies, we included 2 "no-repetition" controls, namely, the many-standards [42] and the cascade sequences [43], in order to account for the possible stimulus-specific effects that could be induced by the oddball paradigm. We found delayed, context-dependent, more robust, and longer-lasting mismatch responses in the rat mPFC than in our previous studies in the rat AC [38,39]. The mismatch responses recorded from both the AC and the mPFC as spiking activity and LFPs correlated in time with the large-scale MMN-like potentials from the rat brain reported in other studies [40,44,45]. Furthermore, the mismatch responses from the mPFC could be mainly identified with PE signaling activity (or genuine deviance detection, in classic MMN terminology), thus confirming their fundamentally different nature from the mismatch responses recorded in the AC.

## Results

In order to find auditory mismatch responses and PEs in the mPFC, we recorded sound-evoked neuronal activity in the secondary motor cortex (M2), the anterior cingulate cortex (ACC), the prelimbic cortex (PL), and the infralimbic cortex (IL) of 33 urethane-anesthetized rats (Fig 1A). For this purpose, we used sets of 10 pure tones arranged in different sequences to create distinctive contextual conditions: the deviant conditions (DEV ascending, DEV descending, and DEV alone) and the standard condition (STD) of the oddball paradigm (Fig 1C), along with their corresponding no-repetition control conditions (CTR), provided by the many-standards (CTR random) and cascade sequences (CTR ascending and CTR descending; Fig 1D).

In the vein of human MMN research [43], we used CTRs to dissociate the higher-order processes of genuine deviance detection or abstract PE signaling from the possible contribution of other lower-order mechanisms related to spectral processing and SSA [21]. On the one hand, CTRs cannot induce SSA or repetition suppression on the auditory-evoked response, in contrast to the STD. On the other hand, CTR patterns remain predictable and should not trigger deviance detection or PE signaling, or at least not as intensely as the DEV [20] (see Oddball paradigm controls for more detailed rationale). By comparing auditory-evoked responses in each condition, we could quantify the estimated contribution of each process to the total mismatch response in the form of 3 indices (Fig 1B): index of neuronal mismatch ($iMM = DEV–STD$), index of repetition suppression ($iRS = CTR–STD$), and index of prediction error ($iPE = DEV–CTR$). Therefore, the iMM quantifies the total mismatch response; the iRS estimates the portion of the mismatch response that can be accounted for by the adaptation hypothesis; and the iPE reveals the component of the mismatch response that can only correspond to genuine deviance detection (according to the sensory-memory hypothesis) or to PE signaling (under a predictive processing interpretation).

In the following sections, we present the results of recording from 83 sound-driven multiunits across all mPFC fields (M2: 25; ACC: 20; PL: 20; IL: 18; Fig 2A), where we were able to test a total of 384 tones at every aforementioned condition (M2: 132; ACC: 90; PL: 81; IL: 81),

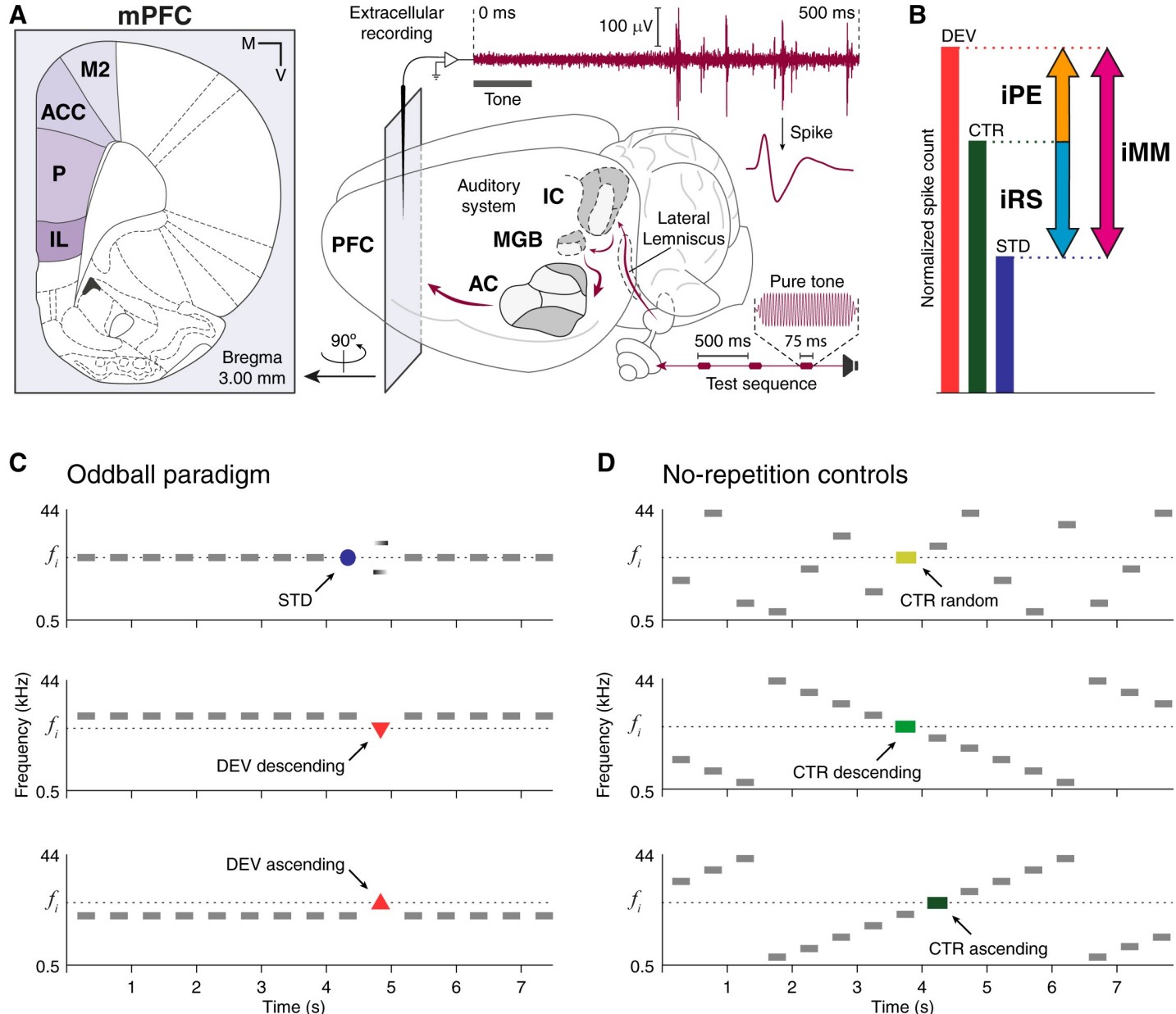

**Fig 1. Experimental design. (A)** Schematic representation of an experimental setup for extracellular recording of auditory-evoked responses in a rat brain. In the left sublet, a schematic coronal section where mPFC fields are highlighted in violet tones. At the right, maroon elements represent the flow of auditory information during the experimental session, from the speaker through the rat brain and into a raw recording trace. **(B)** Decomposition of mismatch responses using the CTR and quantification in 3 indices. **(C)** Three possible experimental conditions within an oddball paradigm for a given tone of interest $f_i$ (colored). **(D)** Three possible control conditions for a given tone of interest $f_i$ (colored). At the top, the many-standards sequence; at the middle and bottom, 2 versions of the cascade sequence. AC, auditory cortex; ACC, anterior cingulate cortex; CTR, control condition; DEV, deviant condition; IC, inferior colliculus; IL, infralimbic cortex; iMM, index of neuronal mismatch; iPE, index of prediction error; iRS, index of repetition suppression; M, medial; MGB, medial geniculate body; mPFC, medial prefrontal cortex; M2, secondary motor cortex; PFC, prefrontal cortex; PL, prelimbic cortex; SEM, standard error of the mean; STD, standard condition; V, ventral.

between 1 and 8 per multiunit (Fig 2C). Although the frequency-response areas (FRAs) appeared unstructured (Fig 2B), these multiunits exhibited robust responses to many combinations of frequency (0.6 to 42.5 kHz) and intensity (25 to 70 dB SPL) during experimental testing (Fig 2C and 2D). This indicates that the auditory sensitivity of mPFC neurons is

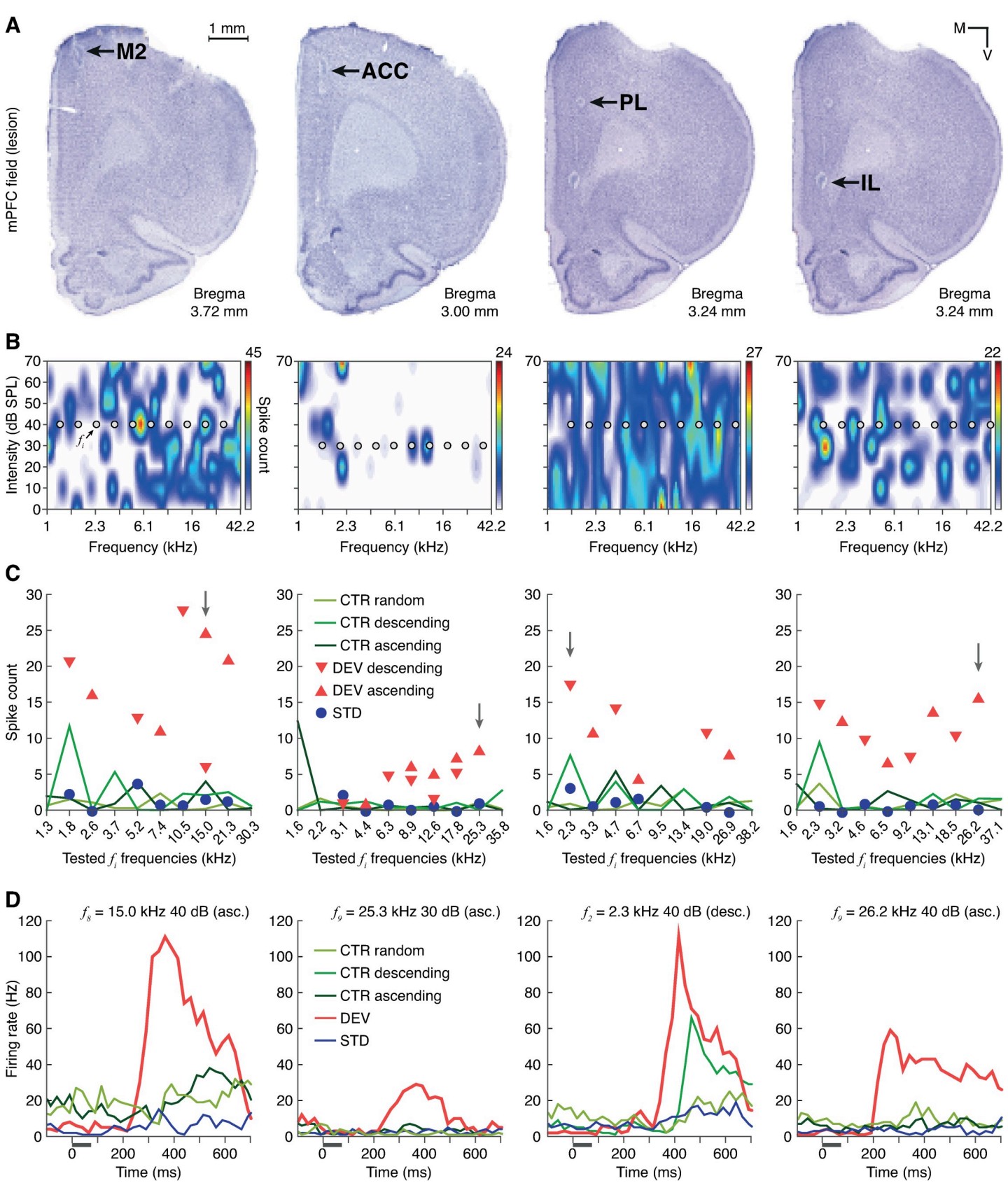

**Fig 2. Multiunit recording examples from each mPFC field. (A)** Coronal mPFC sections where electrolytic lesions (black arrows) mark the recording sites of the multiunits whose auditory-evoked responses are plotted in the sublets below. Hence, column-wise sublets correspond to the same multiunit. **(B)** FRA of 1 multiunit from each mPFC station. Within each FRA, 10 gray dots mark the set of 10 pure $f_i$ tones selected to generate the testing sequences (Fig 1C and 1D), whose evoked response is plotted in the sublet below. **(C)** Multiunit spike counts for every experimental condition of the 10 $f_i$ tested. A vertical gray arrow points at the $f_i$ tone whose peristimulus time histogram is plotted in the sublet below. **(D)** Peristimulus time histogram showing the firing rate elicited by each experimental condition tested for 1 $f_i$ tone, illustrated as a gray horizontal line. The underlying data for this Figure can be found in S1 Data. ACC, anterior cingulate cortex; CTR, control condition; DEV, deviant condition; FRA, frequency response area; IL, infralimbic cortex; mPFC, medial prefrontal cortex; M2, secondary motor cortex; PL, prelimbic cortex; STD, standard condition.

fundamentally driven by the contextual characteristics of auditory stimulation, rather than its spectral properties.

## Context-dependent responses and large PE signals across all mPFC fields

First, we compared the responses elicited by the many-standards and the cascade sequences. Similarly to previous works studying the rat AC [39] and the human MMN [46], we found no significant differences between CTR random, CTR ascending, and CTR descending (Fig 1D), neither within each mPFC field nor for our whole sample (Wilcoxon signed-rank test). Therefore, we used the cascade-evoked responses as CTR for the rest of analyses, based on the theoretical advantages that the cascade sequence offers over the many-standards sequence to control for effects of spectral processing (see Oddball paradigm controls for a detailed rationale) [43].

DEV evoked the most robust discharges across all mPFC fields, usually more than doubling the responses elicited by any other condition (Fig 2C and 2D). Median normalized response to DEV was significantly larger than that to STD or CTR (within-field multiple comparisons Friedman test; Table 1; Fig 3B). Only in M2 the difference in the responses to CTR and STD reached statistical significance ($p = 0.0490$), whereas the distribution of CTR and STD responses proved to be too overlapped in the rest of mPFC fields (within-field multiple comparisons Friedman test; Table 1; Fig 3B). The iMM revealed very large and significant mismatch responses coming from all the mPFC fields (within-field multiple comparisons Friedman test; Table 1; Fig 3C, in magenta). Most of these robust mismatch responses could be accounted for by strong PE signaling, as high iPE values were very significant and very close to those of the iMM (within-field multiple comparisons Friedman test; Table 1; Fig 3C, in orange). Conversely, iRS values were very low in general, and only M2 showed a median iRS significantly different from zero (within-field multiple comparisons Friedman test; Table 1; Fig 3C, in cyan). Remarkably, the values of each index did not differ significantly between mPFC fields (Kruskal–Wallis test with Dunn–Sidak correction; $p > 0.05$ for all comparisons with the 3 indices), so a hierarchical relationship between mPFC fields during the processing of auditory contexts cannot be established in our sample.

According to "standard" implementations of cortical predictive processing [47], error units forwarding PEs are located in superficial layers (II/III), while expectations are encoded by prediction units found in the deep layers (V/VI). Index variations could be expected between superficial and deep mPFC layers, so we attempted to pinpoint the laminar location of our multiunits by means of electrolytic lesions (Fig 2A). Given that such lesions can cover diameters of about 300 μm, half of our multiunit sample had to be excluded from this analysis, as our conservative histological assessment deemed their location inconclusive. Nevertheless, this restrictive histological analysis allowed us to comfortably locate the rest of our multiunit recordings within layers II/III (19 multiunits, 92 tones) or layers V/VI (22 multiunits, 113 tones). Unfortunately, we could not find any significant index changes between II/III and V/VI groups, neither within each mPFC field nor for the whole sample (Wilcoxon signed-rank test).

**Table 1. Median spike counts and indices in each mPFC field.** Significant *p*-values are highlighted.

| | M2 | ACC | PL | IL |
|---|---|---|---|---|
| Number of multiunits | 25 | 20 | 20 | 18 |
| Tested frequencies | 132 | 90 | 81 | 81 |
| *Median raw spike counts* | | | | |
| DEV | 8.6875 | 4.8125 | 6.4750 | 6.0750 |
| STD | 2.7000 | 1.5500 | 1.7750 | 1.1750 |
| CTR | 2.9875 | 1.7000 | 2.5750 | 2.4250 |
| *Median normalized spike counts* | | | | |
| DEV | 0.8693 | 0.8653 | 0.8951 | 0.8511 |
| STD | 0.2751 | 0.2280 | 0.2583 | 0.2202 |
| CTR | 0.3389 | 0.3189 | 0.3225 | 0.3926 |
| *Raw spike count differences, Friedman test* | | | | |
| DEV − STD | 5.9875 | 3.2625 | 4.7000 | 4.9000 |
| *p*-value | **$3.4655 \times 10^{-26}$** | **$2.6737 \times 10^{-14}$** | **$4.5502 \times 10^{-20}$** | **$3.8146 \times 10^{-16}$** |
| DEV − CTR | 5.7000 | 3.1125 | 3.9000 | 3.6500 |
| *p*-value | **$6.9089 \times 10^{-18}$** | **$6.3210 \times 10^{-14}$** | **$6.0892 \times 10^{-14}$** | **$3.8465 \times 10^{-11}$** |
| CTR − STD | 0.2875 | 0.1500 | 0.8000 | 1.250 |
| *p*-value | **0.0490** | 0.9109 | 0.0953 | 0.1249 |
| *Normalized spike count differences, Friedman test* | | | | |
| iMM = DEV − STD | 0.5941 | 0.6373 | 0.6368 | 0.6310 |
| *p*-value | **$3.4655 \times 10^{-26}$** | **$2.6737 \times 10^{-14}$** | **$4.5502 \times 10^{-20}$** | **$3.8146 \times 10^{-16}$** |
| iPE = DEV − CTR | 0.5304 | 0.5464 | 0.5726 | 0.4586 |
| *p*-value | **$6.9089 \times 10^{-18}$** | **$6.3210 \times 10^{-14}$** | **$6.0892 \times 10^{-14}$** | **$3.8465 \times 10^{-11}$** |
| iRS = CTR − STD | 0.0638 | 0.0910 | 0.0642 | 0.1724 |
| *p*-value | **0.0490** | 0.9109 | 0.0953 | 0.1249 |

ACC, anterior cingulate cortex; CTR, control condition; DEV, deviant condition; IL, infralimbic cortex; iMM, index of neuronal mismatch; iPE, index of prediction error; iRS, index of repetition suppression; mPFC, medial prefrontal cortex; M2, secondary motor cortex; PL, prelimbic cortex; STD, standard condition.

## Fast repetition suppression of the response to predictable auditory input

To explore the dynamics of the mismatch responses over time for each mPFC field, we averaged the firing rate to DEV, CTR, and STD in each trial of the sequence across all multiunit recordings. The effect of the position of a stimulus within its sequence is shown in Fig 3D, where each dot indicates the mean response to a given condition, when the position of the trial within the sequence corresponds to the one indicated in the x-axis. We searched for statistical differences between the spike counts of STD and CTR across the trial number. We computed the mean spike counts in groups of 10 trails to obtain 40 measurements (to have the same number of data points for each condition). Then, we calculated the difference of these 40 spike counts to STD minus the 40 spike counts to CTR and tested for statistical significance against zero with a Wilcoxon signed-rank test. Across trial presentation, mean spike counts for STD and CTR events were significantly different only in M2 and IL (Fig 3D; M2: $p = 8.68 \times 10^{-07}$, ACC: $p = 0.87$, PL: $p = 0.14$, IL: $p = 4.87 \times 10^{-06}$).

A power-law model of 3 parameters provided the best fit of the STD responses per mPFC field: $y(t) = at^b + c$ (adjusted R$^2$, M2: 0.358; ACC: 0.259; PL: 0.076; IL: 0.380). Across trials, DEV events maintained a high firing rate (adjusted R$^2$, M2: −0.054; ACC: 0.489; PL: 0.213; IL: −0.054). On the other hand, CTR responses showed repetition suppression, although not as strong and prompt as the STD (adjusted R$^2$, M2: 0.1864; ACC: 0.324; PL: 0.187; IL: 0.245).

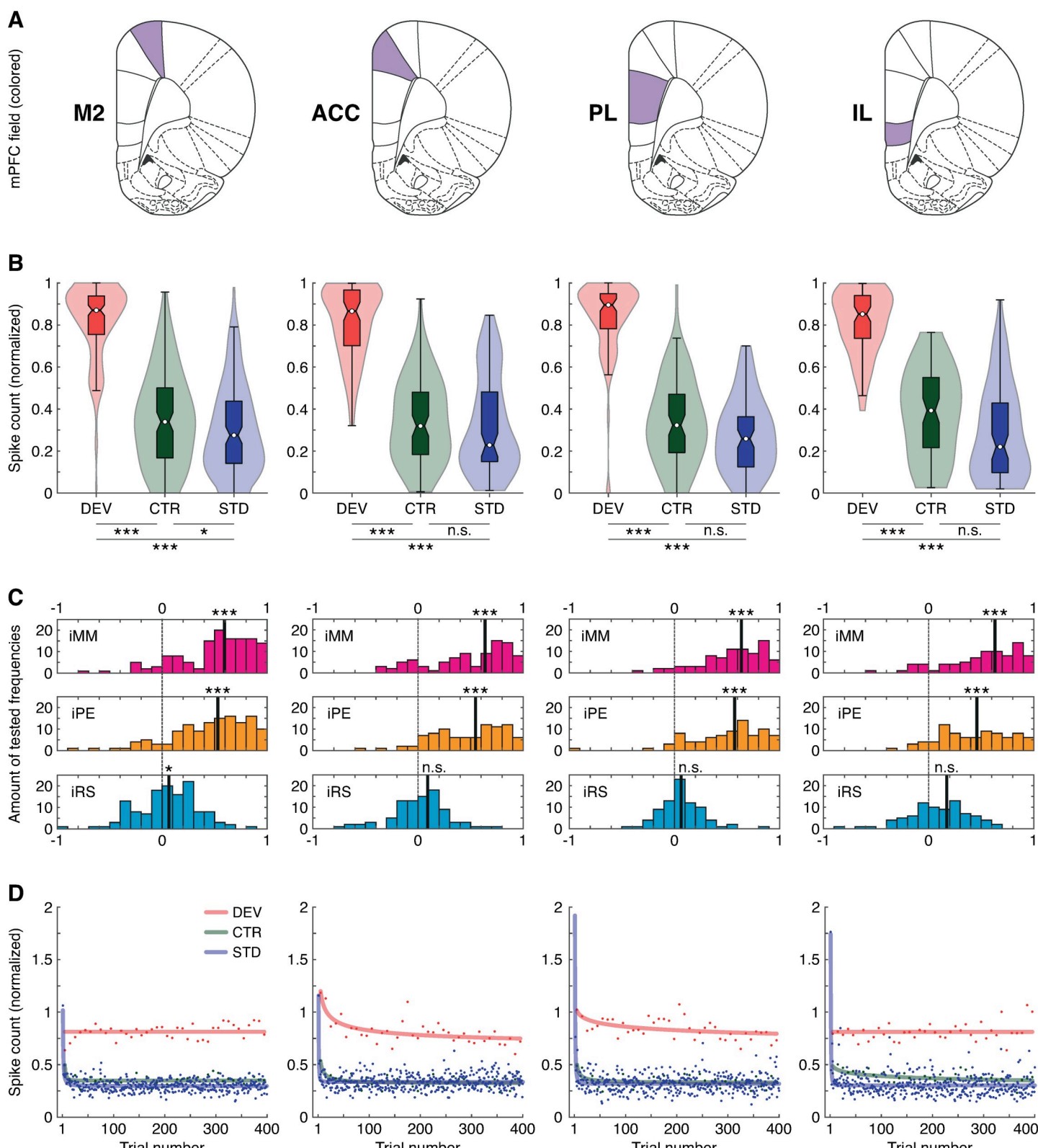

**Fig 3. Spiking activity analysis.** (**A**) Schematic representation of coronal planes highlighting each mPFC field for column-wise reference. (**B**) Violin plots representing the distribution of normalized spike counts for each experimental condition. The boxplots inside each distribution indicates the median as a white dot, the interquartile range as the box, and the confidence interval for the median as the notches. Asterisks denote statistically significant difference between conditions (n.s., nonsignificant,

$^*p < 0.05$, $^{**}p < 0.01$, $^{***}p < 0.001$. **(C)** Distribution of indices in each mPFC field. **(D)** Average spike count per trial number for each condition along the test sequence. Asterisks denote statistical significance against zero (n.s., nonsignificant, $^*p < 0.05$, $^{**}p < 0.01$, $^{***}p < 0.001$). The underlying data for this Figure can be found in S2 Data. AC, auditory cortex; ACC, anterior cingulate cortex; CTR, control condition; DEV, deviant condition; IC, inferior colliculus; IL, infralimbic cortex; iMM, index of neuronal mismatch; iPE, index of prediction error; iRS, index of repetition suppression; MGB, medial geniculate body; M2, secondary motor cortex; PL, prelimbic cortex; STD, standard condition.

Only the repetition suppression to STD manifested very fast and robustly across trials in all mPFC fields ($b$ parameter [with 95% confidence intervals]: M2, −1.373 [−1.656 to −1.089]; ACC, −2.247 [−3.138 to −1.357]; PL, −1.951 [−3.064 to −0.839]; IL, −2.210 [−2.862 to −1.557]). Only 1 repetition sufficed to yield >50% decay of the initial response. Another repetition attenuated the STD response to levels comparable to the steady-state, where the firing rate remained constant until the end of the sequence ($c$ parameter [with 95% confidence intervals]: M2, 0.296 [0.290 to 0.302]; ACC, 0.337 [0.330 to 0.344]; PL, 0.318 [0.309 to 0.326]; IL, 0.302 [0.293 to 0.312]). These findings mean that only 2 repetitions are needed to generate a precise repetition expectation that suppresses this kind of redundancy in the mPFC.

### Microscopic and mesoscopic measurements of PE signals coincide in time

To identify the overall response patterns of each mPFC field, we computed the population temporal dynamics of the average firing rate as normalized spike-density functions. Consistently across all fields, mPFC multiunits exhibited extremely robust and long-lasting firing to DEV (Fig 4B, in red). DEV responses showed very long latencies, needing more than 100 ms poststimulus onset to become discernible from spontaneous activity. Then, DEV firing increased slowly over a course of more than 200 ms before peaking (DEV spike-density function peak latency, M2: 377 ms; ACC: 396 ms; PL: 464 ms; IL: 352 ms). The peak latency in response to DEV stimuli was longer in the PL than in the other mPFC fields (Wilcoxon rank-sum test, PL versus M2: $p = 4.43 \times 10^{-04}$, PL versus ACC: $p = 4.48 \times 10^{-04}$, PL versus IL: $p = 1.50 \times 10^{-04}$; whereas M2 versus ACC: $p = 0.729$, M2 versus IL $p = 0.490$, ACC versus IL $p = 0.756$). This DEV-evoked activity continued in decay, well into the following STD trial of the oddball paradigm. CTR responses tended to follow these same patterns, although with less robust responses and longer latencies (CTR spike-density function peak latency, M2: 516 ms; ACC: 428 ms; PL: 523 ms; IL: 446 ms), such that the response evoked by the previous tone in the cascade sequence is still visible in the current trial (Fig 4B, in green). Finally, the STD did not evoke any robust responses or clear peaks (Fig 4B, in blue).

To analyze PE signaling within each field, we computed the average iPE for each tested tone recorded in 35 time windows of 20 ms width in the range of −50 to 650 ms around tone onset. We tested the indices for significance against zero (Wilcoxon signed-rank test, FDR-corrected for 35 comparisons, $p < 0.05$). iPE started to be significant at 120 ms in the PL, followed by the IL at 140 ms, and later by the M2 and ACC at 180 ms poststimulus onset. In all mPFC fields, iPE signals exceeded half of the index maximum for a sustained length, from about 250 ms poststimulus onset to the end of the analysis window, beyond 600 ms (Fig 4D, in orange).

The extended period of DEV-evoked spiking activity could be the neuronal trace of an updating process of the internal representation by means of PE signals [24,48], as it has been suggested for the human MMN. However, spike responses reflect local activity at the neuron level, whereas the MMN is a large-scale brain potential. One reasonable way of bridging this gap is to probe the correlation between PEs present in the microscopic level with those present within the LFPs [38,39], which constitute the average synaptic activity in local cortical circuits [49]. Hence, we averaged LFP responses for each condition and station (Fig 4C), as well as the difference between DEV and CTR conditions (Fig 4D, in black). We termed this difference as

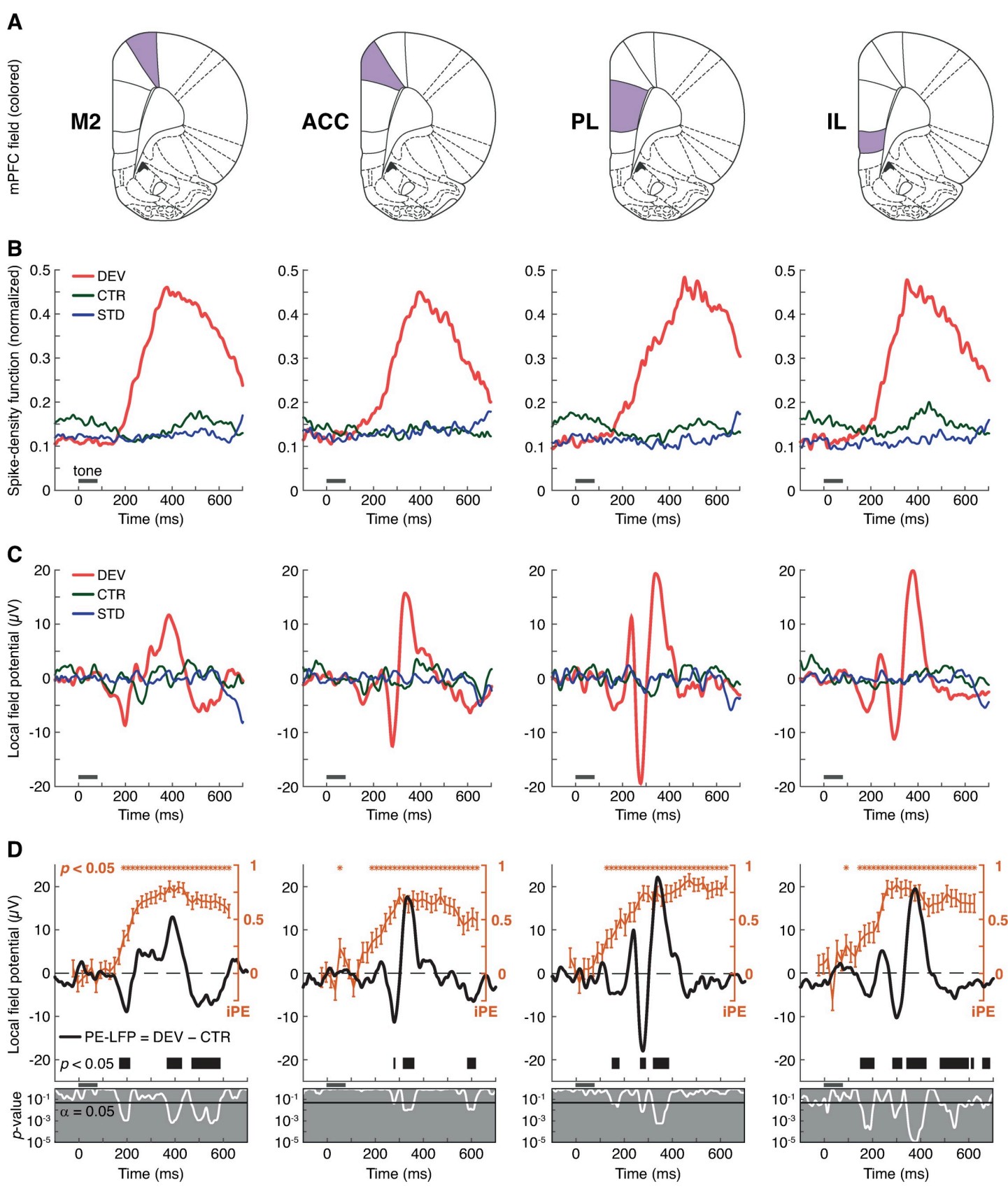

**Fig 4. LFP analysis. (A)** Schematic representation of coronal planes highlighting each mPFC field for column-wise reference. **(B)** Average firing rate profiles of each mPFC field as the normalized spike-density function for every condition. Gray horizontal lines illustrate tone presentation. **(C)** Average LFP across all tested tones and multiunit recordings from each mPFC field for every condition. **(D)** In orange, the time course of the average iPE of the spiking activity (mean ± SEM) where the asterisks above mark a significant iPE value ($p < 0.05$) for the corresponding time window. In black, PE-LFP is the difference wave between the LFPs of DEV and CTR. The thick black horizontal bar below marks the time intervals were the PE-LFP turns significant ($p < 0.05$). The gray sublets below display with a white trace the instantaneous $p$-values corresponding to the PE-LFP of each mPFC field. The underlying data for this Figure can be found in S3 Data. ACC, anterior cingulate cortex; CTR, control condition; DEV, deviant condition; IL, infralimbic cortex; iPE, index of prediction error; LFP, local field potential; mPFC, medial prefrontal cortex; M2, secondary motor cortex; PE-LFP, prediction error potential; PL, prelimbic cortex; STD, standard condition.

"prediction error potential": $PE\text{-}LFP = LFP_{DEV} – LFP_{CTR}$. Indeed, LFP analysis confirmed that the robustness of DEV responses was also clearly observable at the mesoscopic level, in stark contrast to the feeble or nonexistent modulations yielded by CTR and STD (Fig 4C). Significant PE-LFP modulations were also detectable in all mPFC fields, beginning at 147 ms after change onset in IL and PL, followed by M2 at 167 ms and considerably later by ACC at 275 ms (paired $t$ test, FDR-corrected for 428 comparisons, $p < 0.05$; Fig 4D, thick black line). Most remarkably, these PE-LFP modulations occur within the time window where iPE values become significant (Fig 4D, compare the distribution of orange asterisks and thick black lines over time), unveiling a correlation between the PE signals recorded at microscopic and mesoscopic levels.

## Strong responses to unpredictable sounds over a background of silence

In a subset of 9 multiunits (6 rats) from the previously reported data, we tested 39 frequency tones while muting the STD tones of the oddball paradigm, hence obtaining a condition where DEV was presented "alone" (Fig 5A). DEV alone tones were separated by silent periods of a minimum of 1.925 s, equivalent to 3 silenced STD. DEV and DEV alone median spike counts and response patterns did not differ significantly (multiple comparisons Friedman test; Fig 5B and 5C). Although some differences could be observed in the modulations of their LFPs (Fig 5D), these divergencies are negligible as they failed to reach statistical significance (paired $t$ test, FDR-corrected for 428 comparisons; Fig 5E). Thus, the responses of mPFC to unexpected tones are similar, regardless of whether they are presented over a background of silence or interrupting a regular train of other repetitive tones.

## Comparisons between the mPFC and the AC in the rat brain

In order to achieve a more general picture of auditory deviance detection in the rat brain, we also used the data set of a previous work from our lab with similar methodology [39] to study the differences between the mismatch responses in the mPFC and the auditory system. In our previous study, the adaptation hypothesis could only be endorsed in the subcortical lemniscal pathway, whereas predictive activity was identified all along the nonlemniscal pathway and the AC [21,39]. Interestingly, the relative magnitude of mismatch responses along all these auditory centers was comparable, as reflected by their respective median iMM values: 0.49 in the nonlemniscal inferior colliculus (IC), 0.52 in the nonlemniscal medial geniculate body (MGB), 0.50 in the lemniscal (or primary) AC, and 0.60 in the nonlemniscal (or nonprimary) AC. This is also the case in the mPFC, with a median iMM value of 0.59 (Wilcoxon signed-rank test, $p = 6.81 \times 10^{-57}$).

However, the composition of these mismatch responses was fundamentally distinct in the PFC as compared to the auditory system. Repetition suppression was the dominant effect contributing to the mismatch responses of all auditory neurons: 0.46 in both the nonlemniscal IC and MGB, 0.39 in the lemniscal AC, and 0.33 in the nonlemniscal AC. Conversely, the influence of frequency-specific effects in mPFC neurons was almost irrelevant, with a median iRS

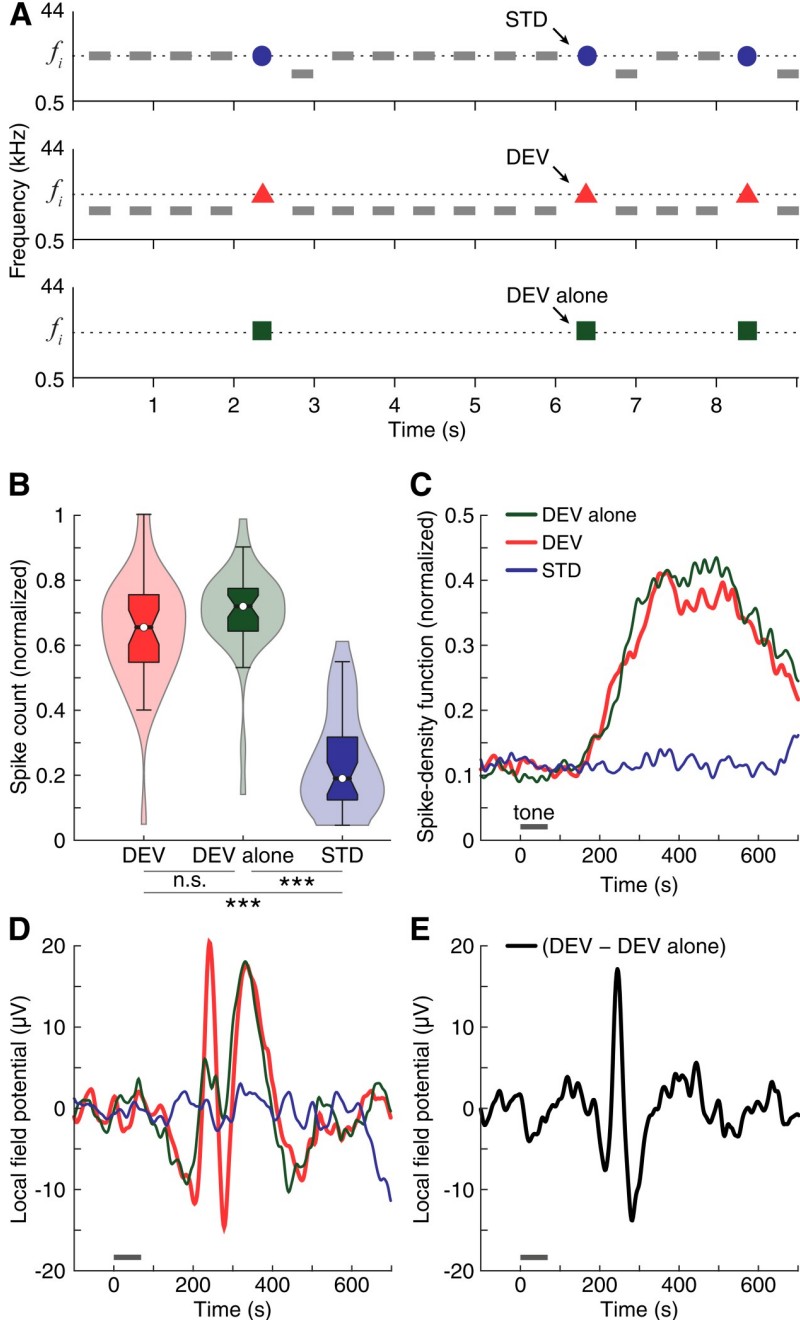

**Fig 5. DEV alone analysis. (A)** Illustration of the DEV alone condition as an oddball paradigm where the STD train is muted. **(B)** Violin plots representing the distribution of normalized spike counts for each experimental condition. The boxplots inside each distribution indicates the median as a white dot, the interquartile range as the box, and the confidence interval for the median as the notches. **(C)** Average firing rate profiles as the normalized spike-density function for every condition. Gray horizontal lines illustrate tone presentation. **(D)** Average LFP across all tested tones and multiunit recording for different conditions. **(E)** Difference wave between the LFP to the DEV and to the DEV alone. The underlying data for this Figure can be found in S4 Data. DEV, deviant condition; LFP, local field potential; STD, standard condition.

value of 0.06 (Wilcoxon signed-rank test, $p = 9.75 \times 10^{-06}$). On the other hand, median iPE values are rather low along the auditory system: 0.03 in the nonlemniscal IC, 0.06 in the nonlemniscal MGB, 0.11 in the lemniscal AC, and 0.27 in the nonlemniscal AC. AC neurons exhibit the most prominent PE signaling, accounting for 22% of the mismatch response in the lemniscal AC and 45% in the nonlemniscal AC. In contrast, PE signaling in mPFC neurons is dominant, with a median iPE value of 0.53 (Wilcoxon signed-rank test, $p = 5.73 \times 10^{-55}$) that accounts for 90% of the total mismatch response (Fig 6A). Thus, spectral properties were the main subject of mismatch responses in the auditory system, while mPFC processing seemed to be abstracted from them.

Statistical comparisons between AC regions and mPFC fields confirmed the general trends described above. The magnitude of the iMM exhibited no significant differences (Kruskal–Wallis test with Dunn–Sidak correction; $p > 0.05$ for all comparisons), but the iPE component grew significantly from the AC to the mPFC (Kruskal–Wallis test with Dunn–Sidak correction; lemniscal AC versus M2: $p = 4.50 \times 10^{-14}$, versus ACC: $p = 1.07 \times 10^{-11}$, versus PL: $p = 4.10 \times 10^{-12}$, versus IL: $p = 1.09 \times 10^{-08}$; nonlemniscal AC versus M2: $p = 3.93 \times 10^{-05}$, versus ACC: $p = 2.12 \times 10^{-04}$, versus PL: $p = 6.74 \times 10^{-05}$, versus IL: $p = 0.011$) to the detriment of iRS, whose proportion drastically shrank to a rather insubstantial contribution to the mismatch response (Kruskal–Wallis test with Dunn–Sidak correction; lemniscal AC versus M2: $p = 1.69 \times 10^{-12}$, versus ACC: $p = 1.11 \times 10^{-12}$, versus PL: $p = 2.61 \times 10^{-10}$, versus IL: $p = 3.12 \times 10^{-06}$; nonlemniscal AC versus M2: $p = 7.46 \times 10^{-08}$, versus ACC: $p = 1.76 \times 10^{-08}$, versus PL: $p = 1.29 \times 10^{-06}$, versus IL: $p = 0.003$). This demonstrates that the nature of mismatch responses in the AC and the PFC is fundamentally different, as predicted by the sensory-memory and the predictive processing hypotheses (Fig 6A).

Temporal dynamics also agree with the abovementioned hypotheses, with the extremely dissimilar latencies observed in the AC and the mPFC point at a sequential processing. Both DEV- and CTR-evoked spiking activity in the AC peaks and stars decaying well before the 75-ms tone has even ended [39]. In stark contrast to the fast AC response, the spiking activity of our whole mPFC multiunit sample began to slowly rise after 150 ms poststimulus onset and took an impressive 462 ms to peak to the DEV and 517 ms to peak to the CTR (Fig 6B). In fact, the entire peristimulus time histogram of a nonlemniscal AC neuron can be represented within the latency of the auditory-evoked responses measured in mPFC neurons (Fig 6C). Regarding the LFPs, an early PE-LFP becomes significant in the AC at about 40 ms and vanishes by 160 ms poststimulus onset, whereas the PE-LFP in our mPFC sample started at 140 ms and lingered with significant magnitudes up to 623 ms poststimulus onset. Both AC and mPFC PE-LFPs coincided precisely with the time course of their respective significant iPE values in spiking activity, thus confirming the PE signaling asynchrony at both microscopic and mesoscopic levels (Fig 6D).

According to data from previous studies in anesthetized rats [38,39], the contrast between AC and mPFC processing is also very apparent in the time needed to explain away STD input. To suppress their initial response to the STD by half, lemniscal AC neurons need 7 repetitions, and nonlemniscal AC neurons 2 repetitions, whereas mPFC neurons only need 1 repetition (Fig 6E, cyan arrow). To reach a steady-state level of maximum attenuation of the auditory-evoked response takes more than the initial 9 STD repetitions in the lemniscal AC, 5 repetitions in the nonlemniscal AC, but only 2 in the mPFC (Fig 6E, dashed lines). This finding rules out the possibility that suppressive effects on the STD could be simply inherited or amplified downstream from the auditory system. On the contrary, the capacity of the mPFC to explain away redundant input more efficiently than the AC supports the predictive processing hypothesis: mPFC expectations are imposed top-down on the AC, thereby influencing earlier stages of auditory processing.

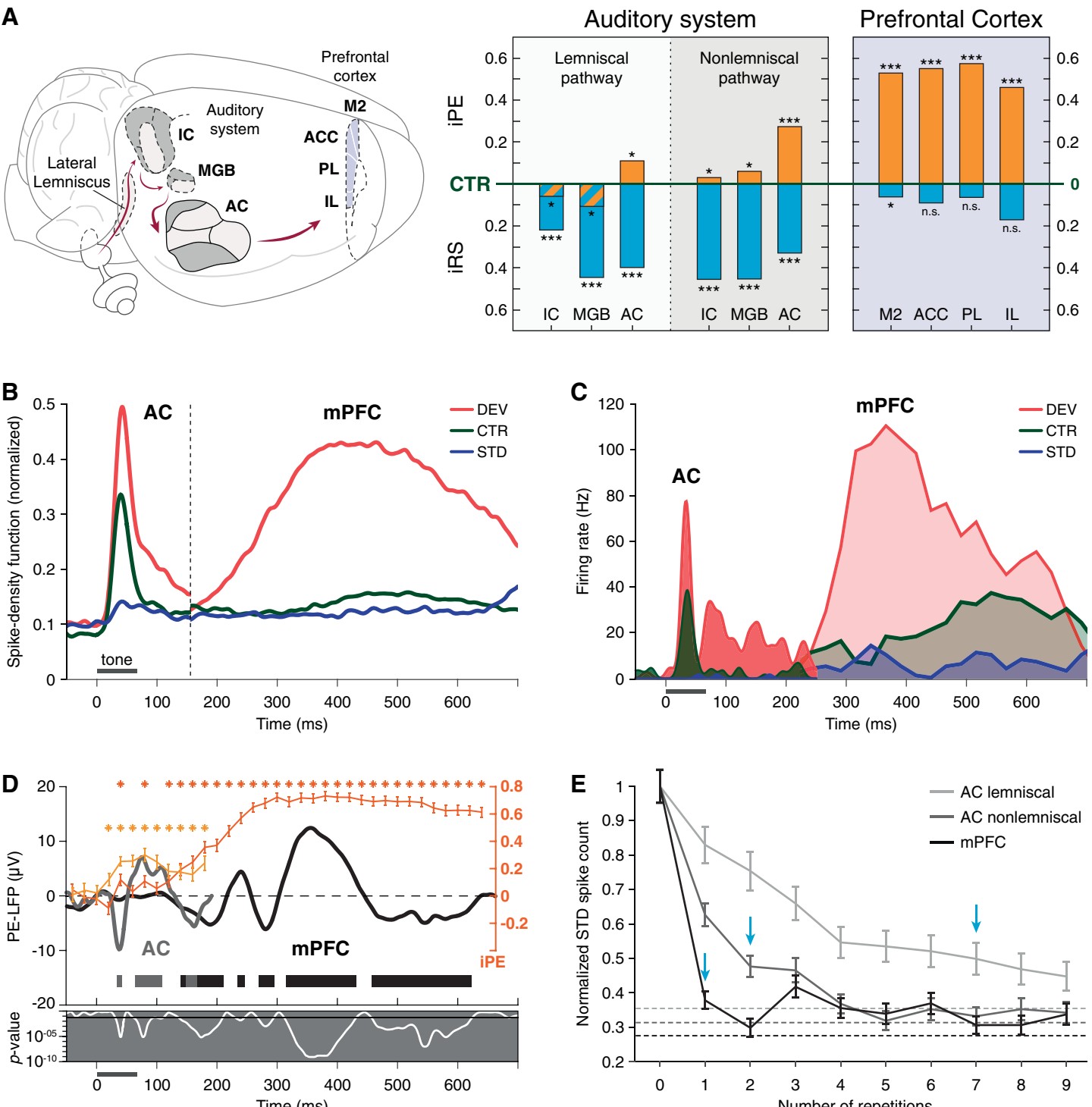

**Fig 6. Comparisons between AC and mPFC responses.** (A) Median iPE (orange) and iRS (cyan) of each auditory or prefrontal subdivision, represented with respect to the baseline set by the CTR. Thereby, iPE is upwards-positive while iRS is downwards-positive (see Fig 1B). Asterisks denote statistical significance of the indices against zero median (n.s., nonsignificant, $^*p < 0.05$, $^{**}p < 0.01$, $^{***}p < 0.001$). (B) Within the interval of 0–150 ms poststimulus onset, average firing rate profile of the nonlemniscal AC as the normalized spike-density function for every condition. Similarly, the mPFC firing rate profile is displayed within the interval of 150–700 ms. Gray horizontal line illustrates tone presentation. (C) Peristimulus time histogram examples of 1 nonlemniscal AC single unit (in solid colors) and 1 mPFC multiunit (in transparent colors), plotted together. Spontaneous activity in the mPFC before 200 ms poststimulus onset has not been represented for clarity. (D) In orange tones, time course of the average iPE of the spiking activity (mean ± SEM) in the nonlemniscal AC (in light orange) and in the mPFC (in dark orange), where the asterisks above mark a significant iPE value ($p < 0.05$) for the corresponding time window. In dark tones, the PE-LFP is the difference wave between the LFP to the DEV and to the CTR

recorded from the nonlemniscal AC (in gray) and from the mPFC (in black). The thick horizontal bar below marks the time intervals were the PE-LFP of the nonlemniscal AC (in gray) and the mPFC (in black) turns significant ($p < 0.05$). The gray sublet below displays the instantaneous $p$-values corresponding to the PE-LFP (in white). **(E)** Average responses for the first 10 STD trials (mean ± SEM) in the lemniscal AC (in light gray), the nonlemniscal AC (in dark gray), and the mPFC (in black). Vertical cyan arrows mark the trial where the initial STD response has undergone more than 50% of attenuation. Dashed lines mark the maximum level of attenuation of the STD response during the sequence (the steady-state parameter of a power-law fit of 3 parameters). The underlying data for this Figure can be found in S5 Data. AC, auditory cortex; ACC, anterior cingulate cortex; CTR, control condition; DEV, deviant condition; IC, inferior colliculus; IL, infralimbic cortex; iPE, index of prediction error; iRS, index of repetition suppression; LFP, local field potential; MGB, medial geniculate body; mPFC, medial prefrontal cortex; M2, secondary motor cortex; PE-LFP, prediction error potential; PL, prelimbic cortex; SEM, standard error of the mean; STD, standard condition.

## Discussion

In this study, we recorded multiunit responses in the rat mPFC to the auditory oddball paradigm and its no-repetition controls, i.e., the many-standards and cascade sequences (Fig 1). We did not observe meaningful differences in the strength of the evoked responses across the 4 mPFC fields or between superficial and deep cortical layers. Unpredictable auditory stimulation prompted robust responses, as compared to the weak (or even absent) activity elicited by sounds that could be expected (Figs 2–5). The time course of the mismatch responses found in the spiking activity and LPFs of the mPFC (Fig 4C and 4D) correlated with that of the frontal sources of the large-scale MMN-like potentials from the rat brain [40,44,45]. Most importantly, our data indicated that mismatch responses of the mPFC are almost purely comprised of PE signaling activity (Figs 3C and 4D), in contrast to the mismatch responses recorded along the auditory system (Fig 6A) [39].

### Unpredictability drives auditory responsiveness in the PFC

Despite the alleged advantages of the cascade over the many-standards sequence for controlling repetition effects during the oddball paradigm [21,43], we did not find any statistically significant differences between the 2 no-repetition controls in the mPFC for the tested parameters. This goes in line with evidence from the auditory system, where the responses evoked by both no-repetition controls were also comparable in AC, MGB, and IC of anaesthetized rats [39]. Such similarity between no-repetition controls tends to be the usual observation in human MMN studies as well [46,50,51]. This suggests that both no-repetition controls are probably processed as a regular succession of pitch alternations, without distinguishing whether those alternations of pitch are random, ascending or descending. Both controls seemingly generate an "alternation expectation" capable of suppressing to a certain extent the auditory-evoked responses in the mPFC, but without inducing stimulus-specific effects of repetition suppression (like STD does). Therefore, the many-standards and the cascade sequences work as largely equivalent CTRs for the oddball paradigm.

Spiking activity in the rat mPFC peaked earlier and higher when evoked by unexpected auditory stimulation, i.e., DEV and DEV alone (which did not differ significantly from each other), more than doubling or even tripling in magnitude the spike response elicited by predictable conditions, i.e., CTR and STD (which only differed significantly from each other in M2; Table 1; Figs 3B, 3D, 4B, 5B and 5C). DEV response dominance was even more pronounced in the LFP analysis, where unexpected DEV and DEV alone conditions prompted robust local field fluctuations whereas the impact of predictable CTR and STD stimulation was negligible (Figs 4C and 5D). We found the same response unbalance between unpredictable and predictable stimulation conditions in all mPFC fields, regardless of whether recordings were performed in superficial or deep cortical layers. The robust mismatch between mPFC responses to unexpected and predictable conditions resulted in similarly high values of iMM (*DEV–STD*) and iPE (*DEV–CTR*). Conversely, the meager or insignificant values of iRS (*CTR–STD*) indicate that the influence of frequency-specific effects is rather irrelevant in the mPFC

(Table 1; Figs 1B, 3C and 6A). Hence, the mismatch responses evoked in the mPFC by the auditory oddball paradigm are better explained as pure PE signaling (for more detailed rationale, see Oddball paradigm controls).

Reports from other frontal sources have found comparable results despite using different methods, recording techniques and model species. Spiking responses in the lateral and ventral orbitofrontal cortex of anesthetized and awake mice also found a great predominance of DEV responses over STD responses [52]. Epidural electrodes placed over the frontal cortices of awake and freely moving rats [40,45] recorded stronger ERPs to DEV than to CTR or STD. In awake macaques, 1 study using multichannel electrodes placed in the dorsolateral PFC found larger responses to DEV than to STD [53], while another using ECoG found strong mismatch responses in the PFC to deviant changes within a roving-standard paradigm, but not to repetitions or the many-standards control [54]. Regarding invasive research in human patients, ECoG studies have consistently proven that, in contrast with the AC, the PFC ceases responding to DEV when its occurrence can be expected [34,37,55]. Although the different prefrontal locations analyzed in the aforementioned studies across rodents, macaques and humans should not be hastily regarded as direct homologs [56], all these works agree in that the key driver of auditory responsiveness in the PFC is unpredictability.

## The neuronal substrate of MMN-like potentials in the rat brain

According to our results, PE spiking activity starts appearing at 120 ms poststimulus onset. About 100 ms later, PE signaling becomes very prominent (iPE >0.5), where it remains more or less sustained beyond 600 ms poststimulus onset, even after the next tone in the sequence has been presented (Figs 4D and 6D, in orange). Most remarkably, such time distribution of the iPE spans enough to include all significant PE-LFP modulations in every mPFC field (Figs 4D and 6D, in black). Therefore, the time course of PE signaling observed in the mPFC at microscopic level coincides in time with that observed at mesoscopic level.

At macroscopic level, ERPs from awake rats exhibited strong mismatch responses beginning about 40 ms poststimulus onset [40,44,45]. Similarly, both our spiking activity and LFP analyses confirmed that early PE signaling starts about 40 ms poststimulus onset in the AC until about 150 ms, when the PFC takes over and continues PE signaling beyond 600 ms poststimulus onset (Fig 6B and 6D). Moreover, the strongest MMN-like potentials are reported in the time window of 100 to 500 ms [40,44,45], precisely coinciding with the period where we registered the most intense PE spiking activity (iPE >0.5), as well as the highest peaks in the PE-LFP (Figs 4D and 6D). Thus, our data allow to correlate the microscopic, mesoscopic, and macroscopic levels at which PE signaling can be detected in the rat PFC. Since the so-called MMN-like potentials are regarded as the rat analog of the human MMN [41], our results could model the possible neuronal substrate of the frontal MMN generators.

## Different nature of PE signaling in the AC and the PFC

Compared to our previous work in the AC [38,39], evoked responses to pure tones in the mPFC were relatively rare and difficult to find. Multiunits that responded to stochastic bursts of white noise during search then exhibited unstructured FRAs, where a concrete receptive field could not possibly be determined (Fig 2B). However, these same multiunits fired consistently in response to many combinations of frequencies and intensities when the tested pure tones were embedded within an experimental sequence (Fig 2C and 2D). Thus, whereas AC processing was clearly driven by the spectral properties of auditory stimulation, auditory sensitivity in mPFC neurons seemed solely dependent on contextual or abstract characteristics. In the same vein, a previous study of spiking activity and LFPs in alert macaques also found

stimulus specificity in the auditory-evoked responses of the AC, but not the dorsolateral PFC [53]. In addition, frequency-specific effects present in the AC within the train of STD or after a DEV were not apparent in the dorsolateral PFC of those alert macaques [53]. Similarly, whereas the iRS in the rat AC can still account for more than half of the mismatch responses [39], at the rat mPFC we found scant or even not significant values of iRS (Fig 6A), thus dismissing any relevant spectral influences in PFC processing.

Our data show that while iMM values in the AC and the mPFC of anesthetized rats are analogous, iPE values are significantly different (Fig 6A). This means that the nature of mismatch responses at the AC is distinct from those at the PFC, despite been paired in their relative magnitude. For this reason, generators at both the AC and the PFC are important contributors to the MMN, but their contributions are fundamentally different in nature, something that has been advocated since the classic sensory-memory interpretation of the human MMN [4,9,10,12] and has also been inherited by the more modern predictive processing framework [20,23]. Given that the iPE can account for 90% of the iMM value, and that in most mPFC fields both indices are not even significantly different, prefrontal mismatch responses can be safely interpreted as genuine deviance detection (in classic terminology) or as pure PE signaling (in predictive processing terminology).

Following this logic, the mPFC would be generating an abstracted mismatch response de novo, signaling "deviance" or a "PE" without reflecting the low-level spectral properties of the driving acoustic stimuli, which have been already represented at earlier processing stages within the auditory system [20,39]. This interpretation is consistent with the huge latency disparities observed between the AC and the mPFC in our anesthetized rats. Whereas AC responses to pure tones take just a few milliseconds to emerge [38,39], evoked responses in the mPFC take hundreds of milliseconds to appear, both at spike activity (Figs 2D, 4B, 5C and 6B) and LFP recordings (Figs 4C, 4D, 5D, 5E and 6D). Prefrontal response delays over 100 ms with respect to the AC have also been reported in the lateral and ventral orbitofrontal cortex of anesthetized and awaked mice [52], as well as in the dorsolateral PFC of alert macaques [53]. Entire AC responses could fit within the latency of the auditory-evoked responses found in the PFC (Fig 6B and 6C). This suggests that AC and PFC processing occur to a certain extent in sequential manner, as described by both the classic sensory-memory [4] and the predictive processing hypotheses [30] of the generation of the MMN. First, acoustic deviances from spectral regularities must be detected at the AC (temporal sources), and only after that, the PFC (frontal sources) can identify global and behaviorally relevant deviations from more abstract internal representations.

Further evidence of the hierarchical relationship between the AC and the PFC could be found in the notable differences between the time each cortical region needs to explain redundant STD input away. According to our previous studies [38,39], neurons in primary or lemniscal AC need 7 repetitions to suppress their initial auditory-evoked response by half, and 2 repetitions in the nonprimary or nonlemniscal AC (Fig 6E, in gray). By contrast, only 1 repetition was enough for the initial auditory-evoked response in the mPFC to drop between >50% and >70%, and a second repetition to reach maximum suppression levels (Fig 6E, in black). Similar suppressive dynamics were reported in the orbitofrontal cortex of anesthetized and awake mice [52], in the dorsolateral PFC of alert macaques [53], as well as in human frontal sources [22].

Given that the PFC responds much later to sound but suppresses redundant auditory input more efficiently than the AC, the mismatch responses observed at the PFC cannot be simply inherited or amplified downstream from the auditory system. The inverse hierarchical arrangement, proposed by the predictive processing hypothesis [30], is thereby more plausible. The PFC is not part of the auditory system; in fact, it is not a sensory processor per se, but

rather an executive center. In more natural conditions, the PFC most likely integrates manifold inputs to generate very complex cross-modality sensorimotor representations [57,58]. These abstract internal representations at the PFC could in turn guide in top-down manner the processing at lower-level systems, hyperparameterizing the more concrete operations carried in their respective (sensory) modalities, and thus increasing overall processing efficiency. In other words, the gestalt acquired at the PFC could be feedbacked to the AC, generating specific expectations in the spectral domain (the native format of AC), but ultimately regarding higher-order properties (such as interstimulus relationships, auditory tokens, or sequence structures) that could have not been computed otherwise in the local AC circuitry. This top-down predictive activity would exert an inhibitory influence on AC responses whenever certain auditory input is already accounted for by the prefrontal gestalt, but any unpredicted information would be conveyed bottom-up in a PE to update the internal representation at the PFC. Thus, hierarchical predictive processing can explain why the PFC exhibits longer latencies than the AC, while also performing more effective and overarching expectation suppression, capable of fully explaining away STD input, and even CTR input. As soon as auditory information becomes redundant to the big picture, it stops reaching the PFC, avoiding cognitive overload, and saving high-order processing resources for more fruitful endeavors.

## Subcortical middle players could relay PE signals to the PFC

Finally, it is worth mentioning that most accounts of deviance detection and PE signaling tend to overrepresent cortical sources, downplaying the role of subcortical contributions. Since the MMN is recorded from the human scalp, the frontotemporal cortical network is more readily accessible for study. The predictive processing framework is also eminently focused on cortical processing [27,47,59]. However, the important contribution of subcortical nuclei is becoming ever clearer in recent literature. Regarding the auditory system, no-repetition controls revealed that SSA could not fully account for the mismatch responses found in the nonlemniscal divisions of the IC and the MGB of anesthetized rats and awake mice. Hence, subcortical auditory nuclei seem to constitute the first levels of the predictive processing hierarchy, which is ultimately responsible for auditory deviance detection [39,60,61].

Human brain research has also identified auditory mismatch signals from subcortical nuclei outside the auditory system, such as the nucleus accumbens [62], the hippocampus [63], or the amygdala [64,65]. Evidence from animal models has been able to confirm these subcortical signals and describe locations and time courses more precisely. Auditory mismatch responses took about 20 ms to appear in the CA1 region of the hippocampus of freely moving mice [66], and 30 to 60 ms to show in the basolateral amygdala of alert macaques [53]. Furthermore, like in the PFC, mismatch responses in the basolateral amygdala did not exhibit stimulus-dependent effects [53]. Minding the different model species, these time delays would place the hippocampus and the amygdala right between the response windows observed in the auditory pathway and those in the PFC.

This could provide a potential explanation for the lack of significant differences between mismatch responses across mPFC fields, despite been quite distinct from each other. The mismatch responses we recorded at the rat mPFC resembled to those recorded at the mouse orbitofrontal cortex [52] and the macaque dorsolateral PFC [53]. It is possible that nonauditory subcortical nuclei such as the hippocampus or the amygdala could compute PEs and then broadcast that signal all over the PFC for further processing and integration. Indeed, a very recent study has demonstrated that the emergence of robust and long-lasting mismatch responses in the mouse orbitofrontal cortex is directly controlled from the nonlemniscal MGB through the basolateral amygdala [52]. Therefore, all these auditory and nonauditory

subcortical nuclei could be fundamental middle players in the automatic process of deviance detection and PE signaling reflected in the MMN. This is a possibility that should be further explored in future studies.

### Limitations

All theoretical implementations of the predictive processing hypothesis assume that expectations and PEs are computed by separated neuronal types distributed across distinct cortical layers, which should result in characteristic laminar profiles [47,59]. Unfortunately, we have not been able to identify any significant response differences between superficial and deep layers of the mPFC, in contrast to what predictive processing models expect. This lack of differences between layers could be due to the unspecific nature of our multiunit measurements. Extracellular recordings can capture the evoked responses of several neurons within a considerable volume of up to hundreds of $\mu m^3$ around the tip of the electrode. The recorded activity does not always allow for spike sorting and waveform analyses to isolate and assign putative neuronal types to the single units contained within one multiunit recording [67], as it was the case in the present study.

Nevertheless, it is worth mentioning that the concrete role of neuronal types and their laminar distribution is still a subject of intense debate within the predictive processing framework. Several possible but conflicting implementations have been proposed [47,68–71], and empirical evidence from human research is mixed (for an in-depth discussion, see [48]). In fact, previous attempts from our lab and others to find a laminar distribution of mismatch responses which fitted the standard implementation of cortical predictive processing [47] also failed in the AC of rats and mice [38,39,66,72]. Therefore, focused research efforts will be needed to disambiguate this issue in the future.

Lastly, the MMN is a notorious obligatory component of the human ERP, remaining persistent in situations where consciousness is absent, such as during sleep [73,74], anesthesia [75,76], or even coma [77,78]. Hence, the fact that we have been able to record very robust mismatch responses in the rat mPFC during anesthesia further strengthens the link between our data and MMN evidence from human research. Moreover, previous studies of mismatch responses in both the auditory system and the PFC of rodents did not find dramatic differences between anesthetized and awake preparations [39,52,79,80]. Notwithstanding, the use of anesthesia is always a limiting factor that must be minded when comparing these data with those obtained from awake preparations, or when trying to extrapolate possible behavioral implications from the conclusions presented in our study.

## Materials and methods

### Ethics statement

All methodological procedures were approved by the Bioethics Committee for Animal Care of the University of Salamanca (USAL-ID-195) and performed in compliance with the standards of the European Convention ETS 123, the European Union Directive 2010/63/EU, and the Spanish Royal Decree 53/2013 for the use of animals in scientific research.

### Surgical procedures

We conducted experiments on 33 female Long-Evans rats aged 9 to 17 weeks with body weights between 200 and 330 g. Rats were anesthetized with urethane (1.9 g/kg, intraperitoneal). To ensure a stable deep anesthetic level, we administered supplementary doses of urethane (approximately 0.5 g/kg, intraperitoneal) when the corneal or pedal withdrawal reflexes

were present. Urethane preserves balanced neural activity better than other anesthetic agents having a modest balanced effect on inhibitory and excitatory synapses [81]. Normal hearing was verified with auditory brainstem responses recorded with subcutaneous needle electrodes, using a RZ6 Multi I/O Processor (Tucker-Davis Technologies, TDT, Alachua, FL, USA) and processed with BioSig software (TDT), using 0.1 ms clicks presented at a rate of 21/s, delivered monaurally to the right ear in 10 dB steps, from 10 to 90 decibels of sound pressure level (dB SPL), using a close-field speaker. Every 10 hours, we administered 0.1 mg/kg of atropine sulfate (subcutaneous), 0.25 mg/kg of dexamethasone (intramuscular), and 5 to 10 ml of glucosaline solution (subcutaneous) to ameliorate the presence of bronchial secretions, brain edema, and prevent dehydration, respectively. Animals were artificially ventilated through a tracheal cannula with monitored expiratory [$CO_2$] and accommodated in a stereotaxic frame with hollow specula to facilitate direct sound delivery to the ears. Rectal temperature was maintained at approximately 37°C with a homeothermic blanket system (Cibertec, Madrid, Spain). We surgically exposed bregma by making an incision in the scalp at the midline and retracting the periosteum. A craniotomy of approximately 3 mm in diameter was performed above the left mPFC and the dura was removed.

### Data acquisition

We recorded multiunit activity to look for evidence of predictive coding signals under acoustic oddball stimulation across fields of the mPFC of the urethane-anesthetized rat: M2, ACC, PL, and IL. The rodent mPFC combines anatomo-electrophysiological elements of the primate dorsolateral PFC and ACC at a rudimentary level [56]. Experiments were conducted in an electrically shielded and sound-attenuating chamber. Recording tracts were orthogonal to the brain surface of the left mPFC: approximately 2.5 to 4.68 mm rostral to bregma, approximately 0.2 to 1.8 mm lateral to the midline, and approximately 0.2 to 4.5 mm dorsoventrally. Therefore, we covered the 4 fields of the mPFC and various cortical layers (II–VI). We performed extracellular neurophysiological recordings with glass-coated tungsten microelectrodes (1.4 to 3.5 MΩ impedance at 1 kHz). We used a piezoelectric micromanipulator (Sensapex, Oulu, Finland) to advance a single electrode and measure the penetration depth. We visualized electrophysiological recordings online with custom software programmed with OpenEx suite (TDT, https://www.tdt.com/component/openex-software-suite/) and MATLAB (MathWorks, https://www.mathworks.com/products/matlab.html). Multiunit activity was extracted automatically by manually setting a unilateral action potential threshold above the background noise as an accurate estimation of neuronal population dynamics [82]. Analog signals were digitized with a RZ6 Multi I/O Processor, a RA16PA Medusa Preamplifier and a ZC16 headstage (TDT) at 97 kHz sampling rate and amplified 251×. Neurophysiological signals for multiunit activity were bandpass filtered between 0.5 and 4.5 kHz using a second order Butterworth filter.

The sound stimuli were generated using the RZ6 Multi I/O Processor (TDT) and custom software programmed with OpenEx Suite (TDT) and MATLAB. Sounds were presented monaurally in a close-field condition to the ear contralateral to the left mPFC, through a custom-made speaker. We calibrated the speaker using a ¼-inch condenser microphone (model 4136, Brüel & Kjær) and a dynamic signal analyzer (Photon+, Brüel & Kjær) to ensure a flat response up to 73 ± 1 dB SPL between 0.5 and 44 kHz, and the second and third signal harmonics were at least 40 dB lower than the fundamental at the loudest output level.

### Oddball paradigm controls

One limitation of the mismatch measurements obtained using the oddball paradigm is that the effects of high-order processes like genuine deviance detection or PE signaling cannot be

distinguished from lower-order spectral-processing effects such as SSA [21,25]. The so-called "no-repetition" controls allow to assess the relative contribution of both higher- and lower-order processes to the overall mismatch response [43]. These CTRs of the auditory oddball paradigm are tone sequences that must meet 3 criteria: (1) to feature the same tone of interest with the same presentation probability as that of the DEV; (2) to induce an equivalent state of refractoriness by presenting the same rate of stimulus per second (which excludes the DEV alone from being considered a proper CTR); and (3) present no recurrent repetition of any individual stimulus, specially the tone of interest, thus ensuring that no SSA is induced during the CTR [20].

Whether the CTR-evoked response exhibited signs of expectation suppression, that could be only explained by high-order regularity encoding or predictive processing, capable of explaining away interstimulus relationships more complex than sheer repetition [21,25]. Hence, we can assess the portion of the mismatch response (*DEV–STD*) that can be attributed to the effects of spectral repetition yielded during the STD train, such as SSA [15,16], by comparing the auditory-evoked responses to DEV and to CTR. When the auditory-evoked response is similar or higher during CTR than in DEV, then the mismatch response can be fully accounted for by repetition suppression, and no higher-order process of deviance detection or PE signaling can be deduced (i.e., DEV $\leq$ CTR; Fig 1B). In other words, this result would provide support for the adaptation hypothesis [17,18] while severely undermining the sensory-memory account [4,10]. Otherwise, a stronger response to DEV than to CTR unveils a component of the mismatch response that can only be explained by a genuine process of deviance detection or PE signaling (i.e., DEV > CTR; Fig 1B).

In order to dissociate the relative contribution of frequency-specific effects from processes of genuine deviance detection or predictive processing, we generated 2 different no-repetition CTRs for our oddball paradigms: the many-standards and cascaded sequences (Fig 1D). The many-standards sequence presents the tone of interest embedded in a random sequence of assorted tones, where each tone shares the same presentation probability as the DEV in the oddball paradigm [42]. However, some authors have argued that this CTR-random is not fully comparable with the oddball paradigm, inasmuch as the disorganized succession of tones never allows to form the memory trace of a proper regularity, nor can it generate high-precision expectations, whereas the STD does. Moreover, the random succession of stimuli might generate small mismatch responses, which would underestimate the contributions of deviance detection or predictive processing in the comparison of DEV against CTR [21,43].

The cascade sequence [43] tries to overcome the alleged caveats of the many-standards sequence by presenting tones in a regular fashion, e.g., in an increasing or a decreasing frequency succession. Thus, the stimulus of interest conforms to a regularity—as opposed to the DEV—, but not a regularity established by repetition and susceptible to undergo SSA—contrary to the STD—, making the cascade sequence a more fitted and less conservative CTR than the many-standards sequence. As an additional advantage, the tone immediately preceding our tone of interest is the same in both oddball and cascaded sequences, since only versions following the same direction are compared (i.e., DEV-ascending versus CTR-ascending, DEV-descending versus CTR-descending). This allows to control for another possible spectral sensitivity, which are responses to a rise or fall in frequency between 2 successive tones. For these reasons, the cascade sequence is regarded as a better CTR for the oddball paradigm [21,43].

## Recording protocol

In search of evoked auditory multiunit responses from the mPFC, we presented stochastic trains of white noise bursts and sinusoidal pure tones of 75 ms duration with 5-ms rise-fall

ramps, varying presentation rate and intensity to avoid possible stimulus-specific effects that could suppress evoked responses.

Once auditory activity was detected, we only used pure tones (also 75 ms duration and 5-ms rise-fall ramps) to record the experimental stimulation protocols. All stimulation sequences ran at 2 stimuli per second. First, a multiunit FRA was computed by randomly presenting pure tones of various frequency and intensity combinations that ranged from 1 to 44 kHz (in 4 to 6 frequency steps/octave) and from 0 to 70 dBs (10 dB steps) with 1 to 3 repetitions per tone. In our previous studies in the auditory system [39,60,61], we selected 10 tones at frequency steps of 0.5 octaves to generate our stimulation paradigms within the receptive field determined by the FRA. However, we could not determine clear receptive fields in the multiunit FRAs of the mPFC, so we had to choose the frequencies and intensity of our test sequences based on our observations during manual search, trying to maximize the auditory-evoked response when possible. Our 400-stimuli test sequences were presented in randomized order leaving periods of >10 min of silence in between to minimize potential long-term habituation effects [83]. All test sequences presented while recording from the same multiunit were delivered at the same intensity, but we varied intensity among the different multiunits of our sample to maximize the auditory-evoked response in each case.

For each multiunit, we used all the 10 preselected tones to generate 3 no-repetition sequences (i.e., the many-standards, cascade ascending, and cascade descending) and pairs of consecutive frequencies (within those 10 tones) to generate oddball sequences. An oddball sequence consisted of a repetitive tone (STD, 90% probability), occasionally replaced by a different tone (DEV, 10% probability) in a pseudorandom manner. The first 10 stimuli of the sequence set the STD, and a minimum of 3 STD tones always preceded each DEV. Oddball sequences were either ascending or descending, depending on whether the DEV tone had a higher or lower frequency than the STD tone, respectively (Fig 1C). Additionally, in a subset of experiments, we muted the STD train to measure the response of the tone of interest over a background of silence, as a DEV alone. The number of test sequences presented to each multiunit depended on the stability of the recording.

### Histological verification

At the end of each experiment, we inflicted electrolytic lesions (10 μA, 10 seconds) through the recording electrode. Animals were afterwards euthanized with a lethal dose of pentobarbital, decapitated, and the brains immediately immersed in a mixture of 4% formaldehyde in 0.1 M PB. After fixation, tissue was cryoprotected in 30% sucrose and sectioned in the coronal plane at 40-μm thickness on a freezing microtome. We stained slices with 0.1% cresyl violet to facilitate identification of cytoarchitectural boundaries (Fig 2A). Histological assessment of the electrolytic lesions to any of the fields of the mPFC was processed blindly to each animal history. Multiunit locations were assigned to M2, ACC, PL, or IL within a rat brain atlas, accordingly with the histological verification and the stereotaxic coordinates in the three axes of recording tracts [84].

### Data analysis

Offline data analyses were performed with MATLAB functions, the Statistics, and Machine Learning toolbox and custom-made MATLAB scripts. We measured multiunit responses to each tested tone by computing a PSTH with 40 trials of every condition (DEV, STD, and CTR). In the case of the STD, we analyzed the last evoked-response before a DEV to have the same number of trials per condition as in DEV and CTR. PSTHs were smoothed with a 6 ms

Gaussian kernel in 1 ms steps to calculate the spike-density function over time (*ksdensity* function). Thereby, we obtained the mean and standard error of the mean (SEM) of spiking rates from −100 to 700 ms around tone onset. The spike-density function of the DEV responses of the mPFC population showed a response latency of approximately 150 ms with a sustained firing spanning up to the next tone (Fig 4B). To avoid overlap of consecutive tone responses, the response analysis window preserved the interstimulus interval of 500 ms and was delayed 100 ms from stimulus onset. For this reason, we did not perform a baseline correction. We only used a baseline window of 50 ms after stimulus onset to assess significantly increased responses to sound to be included in the analyses. We performed a Monte Carlo simulation, which is a probability simulation that withdraws numerical values from several random samplings. We simulated 10,000 PSTHs with a Poisson model of a constant firing rate equivalent to the baseline spontaneous spiking activity and thus, a null distribution of baseline-corrected spike count was generated from the PSTHs. We computed a *p*-value for the original baseline-corrected spike count as $p = (g+1)/(N+1)$, where $g$ is the count of null measures $\geq$ baseline-corrected spike count and $N = 10,000$ is the size of the null sample. The significance level was set at $\alpha = 0.05$.

To compare across different multiunits, we normalized the auditory-evoked responses to each tone of interest in 3 testing conditions as follows:

$$Normalized\ DEV = DEV/N$$

$$Normalized\ STD = STD/N$$

$$Normalized\ CTR = CTR/N$$

where

$$N = \sqrt{DEV^2 + STD^2 + CTR^2}$$

is the Euclidean norm of the vector defined by the DEV, STD, and CTR responses. Thereby, normalized responses are the coordinates of a 3D unit vector defined by the normalized DEV, normalized STD, and normalized CTR responses that ranged between 0 and 1. This normalized vector has an identical direction to the original vector defined by the non-normalized data and equal proportions among the 3 response measurements.

To quantify and facilitate the interpretation of the oddball paradigm controls, we calculated the indices of neuronal mismatch (iMM, computing the overall mismatch response), repetition suppression (iRS, accounting for lower-order frequency-specific effects), and prediction error (iPE, unveiling higher-order deviance detection or PE signaling activity) with the normalized spike counts as:

$$iMM = Normalized\ DEV - Normalized\ STD$$

$$iRS = Normalized\ CTR - Normalized\ STD$$

$$iPE = Normalized\ DEV - Normalized\ CTR$$

Index values ranged between −1 and 1, where

$$iMM = iRS + iPE$$

Lastly, to analyze the emergence of predictive signals around stimulus presentation, we also calculated the average iPE in 35 time windows of 20 ms width from −50 to 650 ms relative to stimulus onset.

For the LFP signal analysis, we filtered the raw recording between 2.2 and 50 Hz (second order Butterworth filter), and then we aligned the recorded wave to the onset of the stimulus for every trial, and computed the mean LFP for every recording site and stimulus condition (DEV, STD, CTR), as well as the "prediction error potential" (PE-LFP = $LFP_{DEV} - LFP_{CTR}$). Then, grand-averages were computed for all conditions, for each auditory station separately. The *p*-value of the grand-averaged PE-LFP was determined for every time point with a 2-tailed *t* test (Bonferroni-corrected for 428 comparisons, with family-wise error rate < 0.05), and we computed the time intervals, where PE-LFP was significantly different from zero.

Our data set was not normally distributed so we used distribution-free (nonparametric) tests. These included the Wilcoxon signed-rank test, Wilcoxon rank-sum test, and Friedman test (for spike counts, normalized responses, indices, and response latencies), as well as the Kruskal–Wallis test with Dunn–Sidak correction for multiple index comparisons between each field from the mPFC and AC. Only the difference wave for the LFPs (PE-LFP) was tested using a *t* test, since each LFP trace is itself an average of 40 waves, and thus approximately normal (according to the Central Limit Theorem). For multiple comparison tests, *p*-values were corrected for false discovery rate (FDR = 0.1) using the Benjamini–Hockberg method [85].

To analyze the time course of suppression over the auditory-evoked response, we measured the DEV, STD, and CTR responses of each tone of interest as average spike counts (each unit normalized to the Euclidean norm, as previously explained) for every trial number within the sequence, for each field separately [38]. Given that the Euclidean Norm vector was calculated for each unit based on the mean DEV, CTR, and STD responses, some individual trials have values above 1. We included all the standard tones, not just the last standard before a deviant event as previously. Thereby, we ordered average normalized spike counts at their absolute trial position within the sequence and generated the time course of responses from the beginning of the sequence. Then, we fitted these time series to various models, namely, linear, exponential, double exponential, inverse polynomial, and power-law with 2 or 3 coefficients. We used the *fit* function in MATLAB that computes the confidence intervals of the fitted parameters and the adjusted $R^2$, the coefficient of determination of the function fit.

For the additional data set including the DEV alone, tests of sound-driven enhanced responses, spike-density functions, spike counts, and normalized responses followed the same previously described analyses. This time, the 3 compared conditions were the DEV alone, DEV, and STD. Since this was an additional experiment to compare the influence of different stimulation contexts on DEV responses, the whole sample was merged along the mPFC.

## Supporting information

**S1 Data. Multiunit recording examples from each mPFC field.** ACC, anterior cingulate cortex; CTR, control condition; DEV, deviant condition; IL, infralimbic cortex; M2, secondary motor cortex; PL, prelimbic cortex; STD, standard condition.
(XLSX)

**S2 Data. Spiking activity analysis.** ACC, anterior cingulate cortex; CTR, control condition; DEV, deviant condition; IL, infralimbic cortex; iMM, index of neuronal mismatch; iPE, index of prediction error; iRS, index of repetition suppression; M2, secondary motor cortex; PL, prelimbic cortex; STD, standard condition.
(XLSX)

**S3 Data. LFP analysis.** ACC, anterior cingulate cortex; CTR, control condition; DEV, deviant condition; IL, infralimbic cortex; iPE, index of prediction error; M2, secondary motor cortex; PE-LFP, prediction error potential; PL, prelimbic cortex; SEM, standard error of the mean;

STD, standard condition.
(XLSX)

**S4 Data. DEV alone analysis.** DEV, deviant condition; LFP, local field potential; SEM, standard error of the mean; STD, standard condition.
(XLSX)

**S5 Data. Comparisons between AC and mPFC responses.** AC, auditory cortex; ACC, anterior cingulate cortex; CTR, control condition; DEV, deviant condition; IC, inferior colliculus; IL, infralimbic cortex; iMM, index of neuronal mismatch; iPE, index of prediction error; iRS, index of repetition suppression; MGB, medial geniculate body; M2, secondary motor cortex; PE-LFP, prediction error potential; PL, prelimbic cortex; SEM, standard error of the mean; STD, standard condition.
(XLSX)

## Acknowledgments

We thank Drs Ryszard Auksztulewicz, Edward L. Bartlett, Conrado A. Bosman, Yves Boubenec, Nell B. Cant, Lucia Melloni, and Kirill V. Nourski for their constructive criticisms and fruitful discussions on previous versions of the manuscript. We also thank Drs Gloria G. Parras and Javier Nieto-Diego for their assistance with neurophysiological recordings and data analyses, as well as Mr. Antonio Rivas Cornejo for the histological processing.

## Author Contributions

**Conceptualization:** Lorena Casado-Román, Guillermo V. Carbajal, David Pérez-González, Manuel S. Malmierca.

**Data curation:** Lorena Casado-Román, Guillermo V. Carbajal.

**Formal analysis:** Lorena Casado-Román, Guillermo V. Carbajal, David Pérez-González.

**Funding acquisition:** Manuel S. Malmierca.

**Investigation:** Lorena Casado-Román.

**Methodology:** David Pérez-González, Manuel S. Malmierca.

**Project administration:** Manuel S. Malmierca.

**Software:** David Pérez-González.

**Supervision:** Manuel S. Malmierca.

**Visualization:** Lorena Casado-Román, Guillermo V. Carbajal.

**Writing – original draft:** Lorena Casado-Román.

**Writing – review & editing:** Lorena Casado-Román, Guillermo V. Carbajal, David Pérez-González, Manuel S. Malmierca.

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
