## [Editor Report · Decision Letter 0]

10 Apr 2020

Dear Manolo, 

Thank you for submitting your manuscript entitled "Prediction errors explain mismatch signals of neurons in the medial prefrontal cortex" for consideration as a Research Article by PLOS Biology. Please accept my apologies for the delay in sending this initial decision to you.

Your manuscript has now been evaluated by the PLOS Biology editorial staff, as well as by an Academic Editor with relevant expertise, and I am writing to let you know that we would like to send your submission out for external peer review.

Please re-submit your manuscript within two working days, i.e. by Apr 14 2020 11:59PM.

Kind regards,

Gabriel Gasque, Ph.D.,

Senior Editor

PLOS Biology

---

## [Decision Letter · Decision Letter 1]

28 May 2020

Dear Manolo,

Thank you very much for submitting your manuscript "Prediction errors explain mismatch signals of neurons in the medial prefrontal cortex" for consideration as a Research Article at PLOS Biology. Your manuscript has been evaluated by the PLOS Biology editors, by an Academic Editor with relevant expertise, and by four independent reviewers. You will note that reviewer 4, Marta Garrido, has signed her comments. 

In light of the reviews (below), we will not be able to accept the current version of the manuscript, but we would welcome re-submission of a much-revised version that takes into account the reviewers' comments. We cannot make any decision about publication until we have seen the revised manuscript and your response to the reviewers' comments. Your revised manuscript is also likely to be sent for further evaluation by the reviewers.

We expect to receive your revised manuscript within 2 months. 

**IMPORTANT - SUBMITTING YOUR REVISION**

Your revisions should address the specific points made by each reviewer. Having discussed these comments with the Academic Editor, we have formulated the following points which you should use as a guide during your revision:

1. We do not expect you to generate new data (experiments), but think that some of the analyses suggested by the reviewers are very pertinent.

2. We agree with reviewer 3’s comment #3 that you should dig deeper into your laminar analysis and compare the different areas of mPFC rather than lumping them together. The ‘canonical circuit’ suggested for hierarchical predictive coding has clear predictions about where predictions error signals should be found (see for instance Keller and Mrsic-Flogel, 2018), as well as separation of feedback and feedforward. We think you have a very interesting dataset which you could potentially exploit more.

3. A related point that reviewer 3 makes is about the lack of specificity. There is very little difference between the responses in the 4 subfields, and in fact in this experiment, we have no example of any area that does not present PE. Since you might have recorded from other areas in this experiment, we urge you to directly compare your results between the mPFC regions and these other regions, if only to make sure that this is not a kind of system-wide response. 

4. You seem to be supporting the hierarchical predictive coding (HPC) model, but the argument in the discussion is based mainly on the amplification of response in mPFC relative to the auditory cortex (compared to data from Parras). However, the hierarchical nature of the HPC (a la Friston and others) is not about amplification of errors from lower to higher levels. It is about higher levels predicting lower levels prediction errors. This should be clarified (see Chao et al., 2018).

5. Another argument for the hierarchical nature of the system is the faster repetition suppression in mPFC (lines 457 to 464). However, the final statement of this paragraph is perplexing: “Hence, the present data is consistent with and further support the hierarchical predictive model where higher areas should encode a regular contextual expectation faster and then send a prediction signal to lower areas to cease responsiveness to subsequent matching inputs”. It is confusing because if indeed mPFC quickly encodes the regularity and suppresses PE signal in lower levels, then why do you find lower levels waiting 5-6 stimuli before suppressing the response?

6. Finally, two reviewers commented on the argument for attention independence. Their perspectives are a bit different, but both correctly note that without manipulation of attention in the experiment, one cannot argue for attention independence. This is not a central argument in the paper anyhow, and we would recommend that you shy away from discussing attention, and rather refer to the stimuli as being task-unrelated, irrelevant, etc, or at least to state simply that being anesthetized, the animals were not selectively attending the stimuli. 

References: 

Chao, Z.C., Takaura, K., Wang, L., Fujii, N., and Dehaene, S. (2018). Large-Scale Cortical Networks for Hierarchical Prediction and Prediction Error in the Primate Brain. Neuron 100, 1252-1266.e1253.

Keller, G.B., and Mrsic-Flogel, T.D. (2018). Predictive Processing: A Canonical Cortical Computation. Neuron 100, 424-435.

------------

Please submit the following files along with your revised manuscript:

*Re-submission Checklist*

*Published Peer Review*

*PLOS Data Policy*

*Blot and Gel Data Policy*

Sincerely,

Gabriel Gasque, Ph.D., 

Senior Editor

PLOS Biology

REVIEWS:

Reviewer #1: I have previously reviewed this paper at a different journal. I am attaching my comments from that review below here, for the benefit of the editor. The authors have done a commendable job in addressing most of my concerns. My one outstanding comment would be related to the writing. It is still unnecessarily confusing and chaotic in many places. Just to illustrate what I mean using a few lines from the abstract:

1. Line 44. "Consistently" should probably read "consistent"

2. Line 45. "High hierarchical areas" should probably read "areas high in the hierarchy" or similar.

3. Line 46. "encode a contextual expectation FAST" - imprecise language. What does fast mean here? The encoding is fast? If so, relative to what? Faster than in other cortical areas? You can just remove the word "fast" from the sentence to improve accuracy of phrasing.

4. Line 46. "send a prediction signal to lower areas to CEASE responsiveness" - combination of grammatical and logical problems. First, this should probably read "suppress" not "cease". Second, the point of the predictive suppression is to suppress responses to predictable stimuli to allow for strong responses to unexpected stimuli, not to "suppress responsiveness", which would imply that all responses are suppressed. 

5. Line 47. The sentence starting with "accordingly" I do not think I understand at all. Again, probably a combination of grammatical and logical problems. "The regularity of representation [...] was fast" what does that mean?

Similarly, throughout other parts of the manuscript. The discussion has much improved but is still quite chaotic and sometimes resembles a list of statements of fact, instead of a discussion. 

If the authors can fix the writing, I would certainly recommend publication. 

The comments I had in the previous review:

In this manuscript Casado-Roman and colleagues describe oddball responses in anterior cingulate, infra- and prelimbic cortex. They find very strong deviant responses consistent with a prediction error and compare these to previous work in auditory cortex. Overall the results are interesting, and if the authors can address the concerns detailed below, I would recommend publication. 

1. The manuscript would likely profit from being condensed. Many of the figures should be relegated to the supplements (e.g. 2,3 & 4). Similarly, in the description of the results, I would focus on the one main finding of the paper (strong oddball responses in fronto-medial cortex). 

2. The authors should more thoroughly discuss what the putative differences are to the Averbeck study (PMID: 30883292) and describe their findings more thoroughly. Statements of the form "However, one study in the PFC of the macaque with deep electrode recording failed to replicate the strong mismatch signals found in primates" are not only grammatically problematic, but also imprecise and do not do the cited work justice. E.g. what is a "deep electrode recording"? Should this read "found in rodents" - the Averbeck study is done in primates. Moreover, they do find deviant responses. Etc. 

3. Performing these experiments under anesthesia is somewhat problematic for an interpretation and should be addressed more carefully. It is clear that MMN can be measured in various behavioral states, but it is difficult to conclude based on this that response in higher order cortical areas are comparable under anesthesia and in the awake animal. The simplest way to address this would be to repeat the experiments in the awake rat, but maybe the authors have a different idea of how to best address this.

Minor

4. The manuscript would profit from being proofread by a native speaker - there are numerous grammatical problems as well as typos.

5. The cortical area the authors refer to as AGm is probably better described as M2 by modern nomencalature.

6. Please include an example of raw recording trace.

7. I would recommend formatting the manuscript in a form that is conducive to reading (e.g. keep figures and figure legends together).

8. Abstract - "Beyond the auditory cortex, the prefrontal cortices integrate error signals to update prediction models." Please rephrase - unclear what is meant (what are "prediction models") and at best this is a theoretical concept not a fact.

9. Abstract last sentence. Please remove or demonstrate how the authors' results relate to a "fundamental mechanism of hierarchical inference" (and explain what that means). 

Reviewer #2: The manuscript entitled "Prediction errors explain mismatch signals of neurons in the medial prefrontal cortex" by Casado-Roman and colleagues continues an important research theme of this lab to the medial prefrontal cortex, namely the detailed analysis of predictive coding in the auditory system. The results themselves are novel and exceptionally salient, the manuscript is well written and addresses most of the questions that came to mind while reading the results. My main concern lies with some of the statistical analyses, and while I think these need to be addressed, I don't think they would substantially alter the results or their interpretation.

As mentioned above, my only significant confusion about the results lies in the statistical analysis in particular the statistics underlying the iRS evaluation. In several figures it appears that the standard responses are significantly smaller than the controls, e.g. 

- grand average in Figure 1C (and others), where SEMs are indicated, which should translate to significant differences, 

- individual responses in Fig.4, comparing the standards to the embedded standards in the controls, there seems to be a significant difference for most areas (I assume that every dot corresponds to a single standard response, which could be described a bit more clearly). 

Clearly, these would lead to smaller iRS values in comparison to the iPE, but still significant. In contrast all your tables and texts indicate that there are no significant differences between the controls and the standards in these conditions (at least when distributed across areas). Of course it could be the case that the statistical power is too low for individual areas (given the comparably low number of multiunits), but it does not seem so from the plots. I would ask the authors to quantify this, also on the across-area level and potentially correct their conclusion about no iRS effect at all, to a small, but significant effect.

Minor points; 

Fig. 1D: The representation of the inset is a bit midleading, since the response seems to come long-time after the stimulus (this clarifies later with the long response latencies in mPFC). Could you indicate the time-scale for this inset, and maybe also show an estimate of response latency already.

Fig.S1: I think it would be nice to integrate Fig.S1 into Fig.1 to immediately clarify the relative spatial location of these areas.

L81: Wouldn't it be more accurate to say: "The predictability of stimuli reduces the PE signals, which has previously been described as repetitions suppression (RS) on a phenomenological level." If so, this would need some more changes, e.g. in lines 85/6.

L306 What does this sentence mean?: Within these conditions, two tones were never repeated and did not undergo the same influence of the repetitive effect as standard events

L.329: why is the rate of RS described as comparable, if there is a two-fold difference between M2 and PL? Form the plots, ACC should have the slowest decrease, which should result in a smaller absolute exponent b.

L335: I find this characterization a bit 'out of context': for the controls, not only the 'standards' but every other stimulus is repeated equally often. So, I think it would be more accurate to also analyze the activation of these stimuli in addition (where you would likely also find that they are equally suppressed, probably because they are not so 'statistically unlikely' (within their respective sequences), compared to the deviants (which you indicated with the words 'when acoustic input matches the current prediction', but this could be more explicit).

Fig. 4: A and B labels missing.

Reviewer #3: Review PBIOLOGY-D-20-00821R1

Summary

Casado-Román and colleagues present a very interesting study about sensory prediction error signals in the Prefrontal Cortex of anaesthetized rats. The goal of this study is to identify the involvement of prefrontal cortices in the generation of prediction error signals and Mismatch Negativity. By presenting different sequences of acoustic stimuli and multi-unit recordings they isolate very strong prediction error (PE) signals in 4 different fields of the medial prefrontal cortex (PrL, IL, ACC and M2). This is certainly by far the clearest and strongest demonstration of a PE signal on the level of multi-unit activity and deserves a lot of credit. This observation is an important confirmation of the model of hierarchical predictive coding in the brain. Therefore, I consider this study an important contribution to the field. However, I think the manuscript in its current form remains too much on a phenomenological level, describing a strong effect with the classical techniques of extracellular electrophysiology. Extending the analysis and probably performing additional experiments may help to provide a deeper insight into the mechanism underlying this very important observation

Major comments

#1 Prefrontal Cortex: I think the authors have to be a bit more careful with their comparison of rodent prefrontal cortex (PFC) and primate PFC. The authors very clearly state in the introduction that previous studies investigating predictive activity in the primate PFC (e.g. Camalier et al. 2019, roughly area 46/8Ad) lacked some critical controls (e.g. see Introduction, line 103-108). The current study is supposed to fill this gap by measuring predictive activity in the rat mPFC with the necessary control conditions to isolate the predictive component (Introduction line 114-116). However, I would argue, that the primate PFC is fundamentally different from the rodent PFC, in particular areas like 46/8 which may have no homologue in rodents at all (e.g. see Wise, 2008: Forward frontal fields: phylogeny and fundamental function or Roberts and Clarke, 2019: Why we need nonhuman primates to study the role of ventromedial prefrontal cortex in the regulation of threat- and reward-elicited responses). Actually, the rodent infra-, prelimbic and cingulate regions are typically compared to primate areas 25, 32 and 24 and not 46 and 8 (where Camalier et al 2019 recorded). I suggest that the authors make it very clear that they are not studying a direct homologue of primate PFC and that the comparison and translation of their results to primate PFC may be difficult and very limited. 

#2 Attention: Throughout the manuscript there are repeated references to attention and the need to study "[…] whether predictive activity independent from attention emerges at the cellular level in the PFC" (Introduction, Line 98-99), see also in the Abstract, line 36-38: "[…] unknown whether prediction error signals are encoded at the neuronal level independently from attention or experience-dependent expectations." and Discussion line 427. I think it is misleading to say that the presented study allows to demonstrate the attention-independency of single/multi-neuron predictive activity. This question cannot be answered with the experiments under anesthesia as presented in the current manuscript and this is acknowledged by authors in the discussion as well (line 393 - 394). I think, this statement should be removed from the Abstract and Introduction or formulated more specific. The use of Urethane as anesthetic is not sufficient to support this view. Only experiments with control over the attentional state of the subject could do so. 

#3 Lack of specificity: Across all areas the authors demonstrate a very strong prediction error signal which is interesting. However, this is a bit surprising as well because the different PFC areas investigate are quite diverse (IL, PrL, ACC and M2). If this strong PE signal indeed is present in all 4 areas and not a result of the unspecific nature of multi-unit recordings, it points to another brain area computing the prediction error and broadcasting it to the PFC areas studied here. The other problem in this study is the that we don't learn anything about the neuronal mechanism behind the observed phenomenon. Again, an increase in specificity may help here, e.g. distinguishing between cell types, layers in individual areas and sub-areas. The authors analyze the layer-specific activity coarsely (Results, line 234-240), however they do so for all areas together.

A better insight into the underlying mechanism may be achieved partially by a more detailed analysis of the existing data (e.g. spike sorting and waveform analysis) or by additional experiments increasing the sample size per layer and area, targeting specific cell-types or inactivating individual (PFC) areas in order to test if the prediction error originates in one of these areas or is indeed almost ubiquitous in the PFC. In the discussion, the authors point towards these approaches (line 486-491) and they should seriously consider to pursue them. 

#4 If the authors indeed measured a single-neuron correlate of MMN, as they suggest, it should be visible in the local field potentials (LFP) as well or at least EEG. It should be pretty straight forward to extract and analyze the LFPs data and check if there is a signature that resembles the human or rat MMN. 

#5: Methods, line 647-649: units had to exhibit a significant increase in firing in at least one of the stimulus conditions (deviant, standard, cascade or many standards, compared to baseline) in order to be included in the analysis. This selection may distort the dataset significantly towards units exhibiting some kind of oddball/PE effect. Therefore, it is important to know if these units are accounted for in table 1 or not (as I assume). If they are not accounted for in table 1, the authors should provide the number of units that were excluded on the basis of this criterium. 

Minor comments

General: Throughout the manuscript, I find the terminology of at "target" tone misleading (e.g. Methods, line 610 and other places). Typically, a target stimulus is used in the context of a behavioral paradigm (e.g. a stimulus to be detected by the subject). However, in the current study no behavior was investigated. 

Abstract, line 35-36: "cortex is postulated to integrate multiple prediction error signals and generate top-down predictions of sensory inputs." The authors assume different prediction error signal generators/sources. But how is this assumption justified?

Introduction, line 80-81: "PE signals are suppressed through a process of repetition suppression (RS)". Is there a reference for this statement?

Introduction, line 92-93: "ECoG reflects the population network dynamics because it records high gamma activity (~80-150 Hz) that reflects multiunit but also synaptic activity." The relationship of high gamma and multi-unit activity in ECoG signals is actually controversially discussed. Apart from that, I don't understand the function of this statement in the context of localizing potential generators of MMN in the human brain and the limited spatial specificity of intra-surgical ECoG mapping. 

Figure 1C: The "Baseline window" (dotted lines) seems to overlap with the preceding "Analysis Window" (dashed lines), including still a period of increased activity from the preceding stimulus (grey lines). How ill this affect the analysis?

Figure 1D: I am not sure I understand what the function of this panel is. It makes sense to show some examples of raw data or individual PSTHs/example units (as in supplementary figure S2). But just a trace of high-pass filtered multi-unit activity is not very informative.

Table 1, Results line 135-138, Methods: If I understand it correctly, for individual units, the authors analyzed all stimuli that evoked a significant response. However, statistically, it seems that all these different stimuli tested even in the same unit are treated as independent, increasing the statistically power massively. But this is not justified when they just come from a small group of neurons, tested with different stimuli. 

Figure 3 legend, line 267: The authors state "[…] mPFC neurons show a long and robust response to deviant events decaying with the next standard tone." (see also Results, line 251-252). This statement is only true when the all units and tested frequencies are lumped together. If one looks at Figure 3C it is obvious that each individual neuron has a peak of activity when Deviant stimuli are presented and that this peak ranges from approx. 100 ms to 700 ms post-stimulus.

Results, line 287-289: "In the mPFC population, latencies of maximum firing rates to deviant events varied notably among frequencies and were uniformly distributed within each field (Fig 3C)." In Figure 3C top row, the distribution of peak latencies seems to be more sigmoid than uniform (at least for M2 and ACC). 

Results, line 342ff and Figure S3: Probably the "inverse" experiment could be interesting as well. Instead of "Deviant-alone" condition (where the deviant "violates" the preceding silence") stimulus omission could be tested. This is one of characteristic features of MMN that even a "no stimulus" event violates are rules and gives rise an MMN wave (e.g. see Yabe et al. 1997). 

Methods, line 545-546: How was the multi-unit activity filtered (type of filter, order, etc.)

Methods, line 550-557: How were the electrolytic lesions performed (current, duration, configuration etc.)?

Methods, line 535: the stereotaxic coordinates have a large variation, approx. 2mm into both the rostro-caudal and the medio-lateral direction. Why was that so? This may significantly reduce the comparability of recording sites between animals and areas as different subareas may have been targeted. 

Methods, line 563: what type of speaker has been used for the auditory stimulation? And what is the definition of a "flat spectrum"? How many dB deviation was accepted?

Methods, line 570-573: It would be interesting to show some tuning curves of the neurons recorded in mPFC, even though the authors state that the tuning seems to be very broad or unclear for pure tones. But in particular under these circumstances it would be important to understand how the individual stimuli were positioned within the tuning curves and how was the frequency of the Deviant stimulus selected?

Methods, line 596: Two stimulus conditions were used for controlling for repetition suppression (RS): many standards and cascade. In order to control for properly for repetition, all stimuli of the control sequences should be within the receptive field of the neuron. Was that the case?

Methods, line 640-641: "We calculated the baseline spontaneous firing rate as the mean firing rate from 0-50 ms during the tone presentation." Why as the spontaneous firing rate computed during the tone presentation and not before the onset of the stimulation? 50 ms is a long time and some signals may have arrived in the PFC already by then. 

Methods, line 707-709: post-hoc / observed "power analysis" was performed on the basis of the effect size iPE. I think that such a post-hoc power analysis does not yield relevant insights and therefore, it can be skipped (e.g. see Zhang et al. 2019: Post hoc power analysis: is it an informative and meaningful analysis?).

Reviewer #4, Marta Garrido: This is a very exciting paper, done in a rigorous manner (with a number of controls), with data and findings that are quite compelling. I fully endorse the publication and only have minor comments that I hope the authors will find useful.

1) Fig 1, C and D - is the recording site mPFC? Can this be specified please?

2) There is a brief justification for why the cascade paradigm is a better control for repetition suppression (RS) than the many-standards. I thought this explanation wasn't clear enough (even the longer description in the methods was unclear) - can you say a bit more?

3) Fig 2 B, can the x labels be added? I assume this corresponds to freq but it's ambiguous.

4) page 14, iPE>iMM, but isn't by definition according to 1B that iMM>=iPE?

5) In the Abstract, Introduction and Discussion the authors mention that it's unclear whether prediction errors are encoded independently of attention. There is actually a growing body of research in humans re modulations of prediction errors with attention, but that's not my point. The point I want to make is that "attention" is perhaps not the best concept to use here and instead "consciousness" would be more appropriate as this relates to the animals being under anaesthesia. There is a subtle but important distinction as even when the animal is conscious, it could still be inattentive. So these data show prediction errors without consciousness (and attention in this case) but these data could not answer the question of whether or not these prediction errors can be modulated by attention. 

6) Fig 5, I thought this would flow better with the results section rather than discussion since this is still about data rather than a proposed model (which could be quite a nice add-on). 

7) In light of Fig 5, the authors propose that there is a progressive mitigation from the IC to AC and eventually vanishing at mPFC. However, I struggle to see that progressiveness - it appears to me more like an abrupt drop. Can this be tested statistically?

8) As above but for iPE in the opposite direction- is it really a progressive build up? Again, can we test this statistically?

---

## [Decision Letter · Decision Letter 2]

6 Nov 2020

Dear Manolo,

Thank you for submitting your revised Research Article entitled "Prediction error signaling explains neuronal mismatch responses in the medial prefrontal cortex" for publication in PLOS Biology. I have now obtained advice from the original reviewers and have discussed their comments with the Academic Editor. You will note that reviewer 2, Bernhard Englitz, and reviewer 4, Marta Garrido, have revealed their identities.

Based on the reviews, we will probably accept this manuscript for publication, assuming that you will modify the manuscript to address the remaining points raised by reviewer 2. Please also make sure to address the data and other policy-related requests noted at the end of this email.

We expect to receive your revised manuscript within two weeks. Your revisions should address the specific points made by each reviewer. In addition to the remaining revisions and before we will be able to formally accept your manuscript and consider it "in press", we also need to ensure that your article conforms to our guidelines. A member of our team will be in touch shortly with a set of requests. As we can't proceed until these requirements are met, your swift response will help prevent delays to publication.

- a cover letter that should detail your responses to any editorial requests, if applicable

*Copyediting*

*Published Peer Review History*

*Early Version*

Sincerely,

Gabriel Gasque, Ph.D.,

Senior Editor,

ggasque@plos.org,

PLOS Biology

DATA POLICY:

Note that we do not require all raw data. Rather, we ask for all individual quantitative observations that underlie the data summarized in the figures and results of your paper. For an example see here: http://www.plosbiology.org/article/info%3Adoi%2F10.1371%2Fjournal.pbio.1001908#s5

These data can be made available in one of the following forms:

Regardless of the method selected, please ensure that you provide the individual numerical values that underlie the summary data displayed in the following figure panels: Figures 3BCD, 4BCD, 5BCDE, and 6ABCDE.

Please also ensure that each figure legend in your manuscript includes information on where the underlying data can be found and that your supplemental data file/s has/have a legend.

Reviewer remarks:

Reviewer #1: The authors have addressed all of my concerns.

Reviewer #2, Bernhard Englitz: The authors have taken great care to improve the manuscript, and most of my points have been addressed.

There is one point, however, regarding the statistical analysis of the data, which still seems odd, and I could not see how the authors' response addressed this point. 

While I agree that in relation to the DEV size (in relation to CTR), the difference between CTR and STD is small, but to the best of my (visual) assessment, should be significant, given the data shown in the previous Figure 4, now shown in Fig. 3D. Let's consider the right panel for region IL. If I understand correctly, the data in this panel are the same as the data in panel B (right), just displayed over time and differently normalized (see below). Is this correct? 

If this is correct, then the distributions of the CTR and STD 'must' be different (of course I cannot assess the precise level of significance), but I think my visual estimate is accurate here. 

In particular if the statistical analysis were to take the location inside the trial sequence into account (i.e. evaluating the difference CTR(iTrial) - STD(iTrial), there is a clear difference between them (which was more easily discernible in the previous figure 4 marker styles), since the dynamics over trials are contributing a large part of the variance. Depending on the number of conditions that went into the Friedman test, I could see how a multiple-testing correction reduced the significance again though.

While I think it is a relatively minor point, I think it is still important to clarify it. Maybe what is shown in Fig. 3B is still different than what is shown in Fig. 3D with respect to the statistical evaluation. You indicate that the data in Figure 3B is Median normalized, but I could not find in the manuscript, what this precisely meant: since the medians in Fig.3B are all different from 0 and 1, it can neither refer to separate divisions/subtractions of the median. Maybe it refers to the overall median, but even then I would expect the whole data to be centered around 0 or 1. Hence, the exact interpretation of this was not clear to me. If one of them is clearly preferable, maybe match the presentation in B and D?

For the benefit of the readers and to withstand all critical future examination I would ask the authors to clarify the procedure for the analysis here, and check in particular whether the STD responses are not infact slightly, but significantly smaller than the CTR responses.

Reviewer #3: The authors addressed all concerns I brought up in the review. In my opinion they did so very successfully and I fully support the publication of the manuscript in Plos Biology now.

Reviewer #4, Marta Garrido: No further comments

---

## [Editor Report · Decision Letter 3]

3 Dec 2020

Dear Dr Malmierca,

On behalf of my colleagues and the Academic Editor, Leon Deouell, I am pleased to inform you that we will be delighted to publish your Research Article in PLOS Biology. 

PRODUCTION PROCESS

Before publication you will see the copyedited word document (within 5 business days) and a PDF proof shortly after that. The copyeditor will be in touch shortly before sending you the copyedited Word document. We will make some revisions at copyediting stage to conform to our general style, and for clarification. When you receive this version you should check and revise it very carefully, including figures, tables, references, and supporting information, because corrections at the next stage (proofs) will be strictly limited to (1) errors in author names or affiliations, (2) errors of scientific fact that would cause misunderstandings to readers, and (3) printer's (introduced) errors. Please return the copyedited file within 2 business days in order to ensure timely delivery of the PDF proof. 

If you are likely to be away when either this document or the proof is sent, please ensure we have contact information of a second person, as we will need you to respond quickly at each point. Given the disruptions resulting from the ongoing COVID-19 pandemic, there may be delays in the production process. We apologise in advance for any inconvenience caused and will do our best to minimize impact as far as possible.

EARLY VERSION

PRESS 

Kind regards,

Erin O'Loughlin

Publishing Editor, 

PLOS Biology

on behalf of

Gabriel Gasque,

Senior Editor

PLOS Biology